# GLC_FCS30D: The first global 30-m land-cover dynamic monitoring product with a fine classification system from 1985 to 2022 using dense time-series Landsat imagery and continuous change-detection method

Xiao Zhang[1, 2], Tingting Zhang[1, 4], Hong Xu[5], Wendi Liu[1, 2, 3], Jinqing Wang[1, 2, 3], Xidong Chen[6] and Liangyun Liu[1, 2, 3, *]

1 International Research Center of Big Data for Sustainable Development Goals, Beijing 100094, China

2 Key Laboratory of Digital Earth Science, Aerospace Information Research Institute, Chinese Academy of Sciences, Beijing 100094, China

3 School of Electronic, Electrical and Communication Engineering, University of Chinese Academy of Sciences, Beijing 100049, China

4 College of Geomatics, Xi'an University of Science and Technology, Xi'an 710054, China

5 The High-Tech Research & Development Center (HTRDC) of the Ministry of Science & Technology, Beijing 100044, China

6 Future Urbanity & Sustainable Environment (FUSE) Lab, The University of Hong Kong, Hong Kong, 999007, China

* Corresponding author Email: liuly@radi.ac.cn

## Abstract

Land cover change has been identified as an important cause or driving force of global climate change and is a significant research topic. Over the past few decades, global land-cover mapping has progressed, however, long time-series global land-cover change monitoring data are still sparse, especially at 30-m resolution. In this study, GLC_FCS30D is described as the novel global 30-m fine land-cover dynamic monitoring dataset, containing 35 land-cover subcategories and covering the period of 1985–2022 with 26 time-steps (maps updated every five years before 2000 and annually after 2000). GLC_FCS30D has been developed using continuous change detection and all available Landsat imagery based on the Google Earth Engine platform. Specifically, we first take advantage of the continuous change-detection model and full time-series Landsat observations to capture the time-points of changed pixels and identify the temporally stable areas. Then, we apply a spatiotemporal refinement method to derive the globally distributed and high-confidence training samples from these temporally stable areas. Next, locally adaptive classification models are used to update the land-cover information for the changed pixels, and a temporal-consistency optimization algorithm is adopted to improve their temporal stability and suppress some false changes. Further, the GLC_FCS30D product is validated using 84,526 globally distributed validation samples in 2020 and achieves an overall accuracy of 80.88% ($\pm$0.27%) for the basic classification system (10 major land-cover types) and 73.04% ($\pm$0.30%) for the LCCS level-1 validation system (17 LCCS land-cover types). Meanwhile, two third-party time-series validation datasets in the United States and Europe Union are also collected for analyzing accuracy variations, and the results show that the GLC_FCS30D offers significant stability for time-series accuracy variation and achieves the mean accuracies of 79.50% ($\pm$0.50%) and 81.91% ($\pm$0.09%) over the two regions. Last, we conclude the global land-cover change information from GLC_FCS30D dataset, namely, the forest and cropland variations dominate global land cover change over past 37 years, and net loss of forests reaches about 2.5 million km$^2$ and net gain in cropland area is approximately 1.3 million km$^2$. Therefore, the novel GLC_FCS30D is an accurate time-series land-cover dynamic monitoring product benefiting from its diverse classification system, high spatial resolution

and the long time span of 1985–2022, thus, it will effectively support global climate change research and promote sustainable development analysis. The GLC_FCS30D datasets are available via https://doi.org/10.5281/zenodo.8239305 (Liu et al., 2023).

**Keywords**: GLC_FCS30D, 1985-2022, Land-cover change, Landsat, change detection, Google Earth Engine

## 1. Introduction

Land cover data are important and necessary for supporting sustainable development goals, maintaining biodiversity, and monitoring natural resources (Liu et al., 2021b; Potapov et al., 2022). The land cover changes directly or indirectly influence the global climate patterns and the speed and magnitude of climate change (Song et al., 2018) and increasingly affects biogeochemical cycles, the carbon cycle, and Earth's energy balance (Foley et al., 2005; Hong et al., 2021; Winkler et al., 2021). Since the industrial revolution, under the dual pressure of global climate change and human activities, global land cover has undergone drastic changes. According to a Global Carbon Project report in 2020, since the industrialization period, land cover and land use changes have contributed to approximately 25% of global greenhouse gas emissions (Friedlingstein et al., 2020), and this trend is exacerbated with the ongoing increase in population and per capita energy consumption (Xian et al., 2022). Therefore, understanding and studying land cover changes are of vital significance for addressing global environmental changes, promoting sustainable development, and safeguarding the Earth's ecological environment.

Remote-sensing techniques, with periodic Earth observation capability and archived massive long-term observation data since 1972, provide the most cost-effective and practical solutions for long time-series large-area land-cover change monitoring. In the past few decades, with the continuous improvement of remote sensing technology and storage and computing capabilities, global land-cover change monitoring (GLCCM) has transitioned from 1-km spatial resolution to fine resolution of 30-m or 10-m and from single-phase mapping to long-term monitoring (Ban et al., 2015; Friedl et al., 2010; Friedl et al., 2022; Giri et al., 2013). In the early stage, GLCCM mainly relied on the time-series MODIS, AVHRR, and Project for Onboard Autonomy (PROBA)-V imagery (Buchhorn et al., 2020; Friedl et al., 2010); for example, Sulla-Menashe et al. (2019) generated a global 500-m annual land-cover products (MCD12Q1) from 2001 to present using time-series MODIS imagery with an overall accuracy of 73.6%. Defourny et al. (2018) integrated time-series PROBA-V and Medium Resolution Imaging Spectrometer (MERIS) observations to develop a global 300-m annual land-cover dynamic dataset (CCI_LC) from 1992 to 2020 with an overall accuracy of 71.5%. These coarse land-cover change products comprehensively captured the spatial patterns of various land-cover types and quantified the global land-cover changes caused by human and natural activities. However, they still had major limitations especially in regions with intense human activity and high spatial heterogeneity because these broken and heterogeneous land-cover changes cannot be captured by coarse-resolution satellite observations (Hansen et al., 2013; Herold et al., 2008; Liu et al., 2021b; Zhang et al., 2021c).

Recently, benefitting from the free access to fine-resolution satellite imagery and powerful computing and storage capabilities, especially after the rise of cloud computing [such as Google Earth Engine (Gorelick et al., 2017) and Microsoft Planetary Computer], fine-resolution land-cover dynamic monitoring is experiencing rapid development. Correspondingly, numerous national and global 30-m land-cover dynamic products have been developed (Chen et al., 2015; Homer et al., 2020; Liu et al., 2021a; Potapov et al., 2022; Yang and Huang, 2021; Zhang et al., 2022). For example, Yang and Huang (2021) used China's land-use/cover datasets (CLUDs) as

the prior dataset and then combined multitemporal classification and spatiotemporal consistency post-processing method to develop an annual 30-m land cover dataset (CLCD) for China from 1990 to 2019. Similarly, Liu et al. (2021a) combined pixel-based classification and spatiotemporal post-processing method to generate the first global 30-m land-cover change products. However, many studies have demonstrated that multiperiod independent classifications lead to significant classification error accumulation in time-series land-cover change monitoring (Sulla-Menashe et al., 2019; Zhu, 2017). For example, Xian et al. (2022) stated that the independent classification strategy suffered the constraints of post-processing requirements, for ensuring the temporal consistency of land-cover change maps. Therefore, although GLCCM has progressed significantly over the past few decades, an accurate global 30-m land-cover change-detection product generated by an efficient land-cover change method is still urgently required.

One of the greatest challenges for large-area land-cover change detection is to select the optimal algorithm to capture the land-cover changes from time-series observations (Healey et al., 2018; Zhu, 2017). Over past few decades, a series of change-detection algorithms have been proposed for monitoring forest disturbance (Huang et al., 2009; Jin et al., 2023; Kennedy et al., 2007; Kennedy et al., 2010; Qin et al., 2021), urban expansion (Liu et al., 2019; Zhang et al., 2021a), cropland dynamics (Dong et al., 2015; Potapov et al., 2021), and land-cover changes (Bullock et al., 2019; Jin et al., 2017; Verbesselt et al., 2010; Zhu et al., 2019). However, most of them were only suitable for regional land-cover change monitoring and some of the algorithms needed prior knowledge (such as for urban expansion). Zhu (2017) systematically reviewed the performances and limitations of various change-detection methods based on multitemporal satellite data, and further explained that the high temporal frequency and multivariate change-detection algorithms are more suitable for large-area and long time-series land-cover changes after solving a problem with a huge amount of computation. Similarly, Xian et al. (2022) and Liu et al. (2019) concluded that dense and continuous change-detection methods had higher accuracy and more robustness than traditional change-detection methods for capturing multiple, complicated changes.

Continuous Change Detection and Classification (CCDC) algorithm, a classical change-detection method based on dense time-series observations proposed by (Zhu and Woodcock, 2014b), is widely used for regional and national land-cover monitoring (Xian et al., 2022; Xie et al., 2022). It used all available Landsat observations to build time-series regression models and then captured the outliers by analyzing the differences between the actual observations and model estimations. Zhu and Woodcock (2014b) demonstrated that the CCDC algorithm attained a general accuracy of 90% and temporal accuracy of 80% for capturing land-cover changes. Thus, it has been adopted by the United States Geological Survey (USGS) as the official algorithm for monitoring land-cover changes over the United States (Pengra et al., 2016). For example, Xian et al. (2022) implemented the CCDC algorithm and all available Landsat data to develop annual land-cover change products over the contiguous United States (CONUS) for 1985–2017 with an overall accuracy of 82.5%.

In summary, over the past decades, land-cover mapping and monitoring has made significant progress; however, time-series global 30-m land-cover dynamic products derived from change-detection algorithms are still lacking. In this study, we had the following three aims: 1) use the continuous change-detection algorithm and full time-series Landsat observations to generate the first global 30-m land-cover dynamic products with fine classification system (GLC_FCS30D) from 1985 to 2022, which contains 35 fine land-cover subcategories with 26 time-steps (maps updating every five years before 2000 and annually after 2000). It should be noted that the GLC_FCS30D updated every five-years before 2000 due to the sparse availability of Landsat 5 imagery, thus, we combine the satellite observations from two years before and after the nominal center year from 1985

to 1995 for ensuring the mapping accuracy of GLC_FCS30D before 2000. 2) Quantifying the land-cover changes and analyze the spatiotemporal change patterns of various land-cover types based on the developed GLC_FCS30D dataset; and 3) quantitatively analyzing the performance of the GLC_FCS30D product using multisourced validation datasets.

## 2. Datasets

### 2.1 Continuous Landsat imagery from 1984 to 2022

All available Landsat imagery from 1984 to 2022, covering Landsat 5 Thematic Mapper (TM), Landsat 7 Enhanced Thematic Mapper Plus (ETM+), Landsat 8 Operational Land Imager (OLI), and Landsat 9 OLI missions, was collected via the GEE cloud-computing platform. Specific measures were taken to build a high-quality continuous time-series Landsat collection. First, all Landsat images underwent atmospheric correction to convert them to surface reflectance using the Landsat Ecosystem Disturbance Adaptive Processing System (LEDAPS) and Land Surface Reflectance Code (LaSRC) methods (Vermote, 2007; Vermote and Kotchenova, 2008). Then, although the Landsat 5, 7, 8, and 9 missions share similar spectral bands, the wavelength differences between the TM, ETM+ and OLI cannot be ignored. Relative radiometric normalization was applied to the TM and ETM+ imagery using the transformation coefficients suggested by Roy et al. (2016).

### 2.2 Global land-cover dataset at 30 m for the year of 2020

The global 30-m land-cover product with fine classification system in 2020 (GLC_FCS30-2020), is the baseline for generating training samples and identifying land-cover information in the temporally stable regions in Section 3. The GLC_FCS30 dataset was developed using locally adaptive classifications and confident and globally distributed training samples in their developed global spatiotemporal spectra library, and then validated to reach an overall accuracy of 82.5% with the basic validation system (Zhang et al., 2021b). Cross-comparisons with other land-cover products showed obvious advantages for the GLC_FCS30 in mapping accuracy and diversity of land-cover types. The GLC_FCS30-2020 dataset is freely available at https://doi.org/10.5281/zenodo.4280923 (Liu et al., 2020).

### 2.3 Global impervious surface dynamic dataset at 30 m from 1985 to 2022

Many studies found that high spatiotemporal heterogeneity and broken constructs of impervious surface caused high uncertainty and difficulty in monitoring their dynamics (Gong et al., 2019a; Zhang et al., 2022), and change detection methods also face issues of both missed detections and false alarms when applied to the dynamic monitoring of heterogeneous impervious surfaces. Thus, we independently produced a time-series global impervious surface dynamic dataset at 30 m (GISD30) during 1985–2022 and then overlaid this thematic dataset on the GLC_FCS30D dataset to ensure their high confidence in the impervious surface dynamics. The GISD30 dataset was developed by combining the sample migration, spectra generalization and local adaptive modeling methods, and then optimized by the spatiotemporal-consistency correction method (Zhang et al., 2022). It was validated to attain the mean overall accuracy of 90.1% over the globe and to perform better than other similar products in capturing the changes in impervious surfaces over time and across different types of landscapes. At the same time, third-party validation also indicated that GISD30 exhibited superior performance among similar global 30-meter impervious surface products (Wang et al., 2022).

### 2.4 Global 30-m wetland datasets from 1985 to 2022

Like the impervious surface, the global wetland dynamic dataset is independently produced because the reflectance spectra of the wetland and phenological variations changed daily with the water levels. The continuous change-detection method would suffer serious commission and omission errors for wetland dynamic monitoring (Gallant, 2015). In this study, the GWL_FCS30 (global 30-m wetland map with a fine classification system) wetland dataset from 1985 to 2022, developed by integrating the automatic sample extraction method, a stratified classification strategy, and time-series Landsat observations (Zhang et al., 2023), would be superimposed on the GLC_FCS30D land-cover dynamic dataset. The GWL_FCS30 was quantitatively assessed to the mean overall accuracy of 86.44% using 25708 validation points, and demonstrated the highest level of performance among other wetland products when it came to capturing the spatial patterns of wetlands during cross-comparisons (Zhang et al., 2023). GWL_FCS30 further splits the wetland into seven wetland subcategories (four inland and three coastal subcategories), and would be overlaid directly onto the GLC_FCS30D dataset not only improves the monitoring accuracy of wetland but also riches the number of land cover types (in Table 1).

## 2.5 Validation datasets

To comprehensively analyze the accuracy metrics for the GLC_FCS30D dataset, two types of validation datasets were collected, including: an independent global validation dataset from 2020 and two third-party time-series validation datasets for the United States and the European Union.

### 2.5.1 Global validation dataset

A total of 84,526 globally distributed validation samples are collected to analyze the accuracy metrics for the GLC_FCS30D dataset in 2020, and their spatial distributions are illustrated in Figure 1. Intuitively, the spatial patterns of the global validation dataset are consistent with the actual global land-cover situation. Specifically, to ensure the confidence and rationality of the validation datasets, several measures are taken, which were explained in detail in our previous work (Zhao et al. (2023). First, a stratified random sampling method is applied by combining the landscape heterogeneity, population density data and Köppen climate groups, which effectively increases the sample size in the heterogeneous landscapes and for these rare land-cover types. Second, for each validation sample, the land-cover type is determined through independent interpretation by trained interpreters after combining high-resolution aerial photography, multitemporal Landsat images, and other relevant ancillary datasets (such as: vegetation coverage, tree height, phenological curves, , and terrain characteristics). Independent interpretation software has also been developed based on the GEE platform (https://eliza-ting.users.earthengine.app/view/crd-vit) for efficiently recognizing the land cover types of each sample. Third, a quality-controlled operation, based on duplicate interpretations, is further used to ensure the confidence level of each validation sample. Each sample is independently labeled by three junior interpreters and then double-checked by the senior experts, and validation samples with huge disparities would be discarded. In addition, because the impervious surface and wetland datasets have been independently produced and validated in our previous works (Zhang et al., 2023; Zhang et al., 2022), the corresponding high-quality validation samples of these two thematic types are also imported to the global validation datasets.

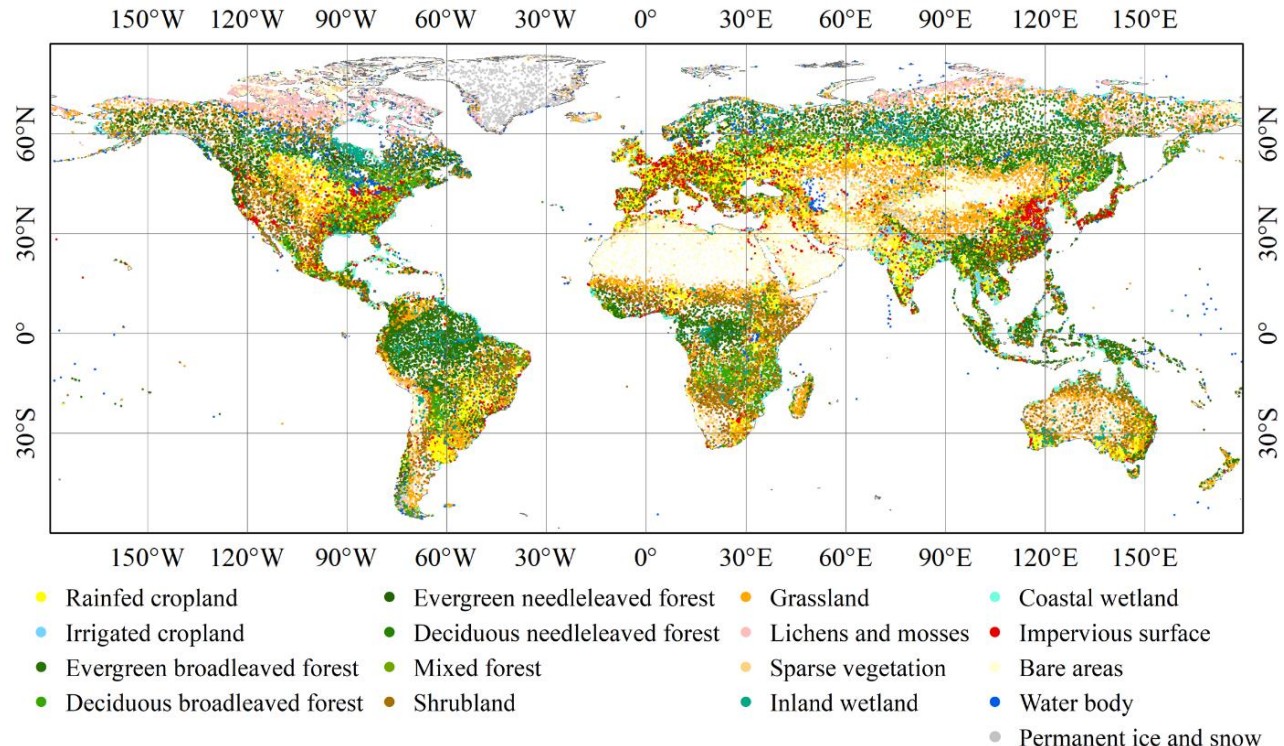

**Figure 1.** Spatial distribution of 84,526 global validation points containing 17 fine land-cover types in the normal year of 2020.

### 2.5.2 Third-party regional time-series validation datasets

Due to the great difficulty in the collection of long time-series global validation datasets, we used two third-party regional datasets for the CONUS and the European Union. The first time-series validation dataset assessed the performance of the Land Cover Monitoring, Assessment, and Projection (LCMAP) Collection 1.0 annual land cover products (Stehman et al., 2021) (called LCMAP_Val, https://www.usgs.gov/special-topics/lcmap/validation-data). The LCMAP_Val consisted of 24,971 validation samples with 30-m spatial resolution and covered time period of 1985 to 2017. It was developed by combining a simple random sampling method and visual interpretation from high-resolution aerial photography, multitemporal Landsat images, as well as other auxiliary datasets. Meanwhile, to guarantee the reliability of each validation sample, the TimeSync auxiliary tool was also adopted to capture the land-cover changes (Stehman et al., 2021). Quality control was implemented through duplicate interpretations (Xian et al., 2022).

The second regional validation dataset was the Land Use/Cover Area frame Survey (LUCAS), which is the most comprehensive and largest land-cover validation dataset in the European Union and is freely available at https://land.copernicus.eu/imagery-in-situ/lucas. It contains 1,090,863 validation points based on a systematic 2 km × 2 km grid and covers the period of 2006 to 2018 with an interval of 3 years (d'Andrimont et al., 2020). Five LUCAS surveys in 2006, 2009, 2012, 2015, and 2018 assessed the time-series accuracies of the GLC_FCS30D. The LUCAS is developed from a combination of field observations and photo interpretation (Büttner and Eiselt, 2013; Ballin et al., 2018); thus, it performs with high confidence and also attracts widespread attention in land-cover validations (Gao et al., 2020; Venter et al., 2022).

### 3. Methods

Figure 2 presents a detailed flowchart for monitoring land-cover changes by combining the Continuous Change Detection (CCD) algorithm, proposed by Zhu and Woodcock (2014b), and the local adaptive updating method. Specifically, the flowchart contains four main procedures including: (1) detecting the temporally stable pixels and the time-points of abrupt changes in these changed land-cover pixels from the continuous change-detection model; (2) deriving the spatiotemporally stable training samples by using the spatiotemporal refinement method from GLC_FCS30 land-cover products and temporally stable masks; (3) building the local adaptive classification models for each local region and then updating the land-cover information in the changed pixels; and (4) using the spatiotemporal consistency optimization method to improve the quality of land-cover change maps and suppress these false changes.

Before detecting the land-cover changed pixels, all 'poor quality' pixels (cloud, shadow and saturated pixels, as well as the Scan Line Corrector Off pixels in Landsat 7) in the continuous time-series Landsat imagery were firstly masked using the CFmask algorithm, which was demonstrated to achieve the overall accuracy of 96.4% and was adopted by the USGS as official cloud- and shadow detection algorithm (Zhu et al., 2015; Zhu and Woodcock, 2012). Then, in terms of these residual cloud pixels (light cloud and haze contaminated pixels), the Tmask (multiTemporal mask) algorithm, which used the temporal information from these clear-sky pixels to improve the cloud-detection capability (Zhu and Woodcock, 2014a), was used to mask the residual cloud pixels. It should be noted that the Tmask has been integrated into the CCD algorithm on the GEE platform as *ee.Algorithms.TemporalSegmentation.Ccdc()*, that is, the effect of 'poor-quality' pixels were minimized.

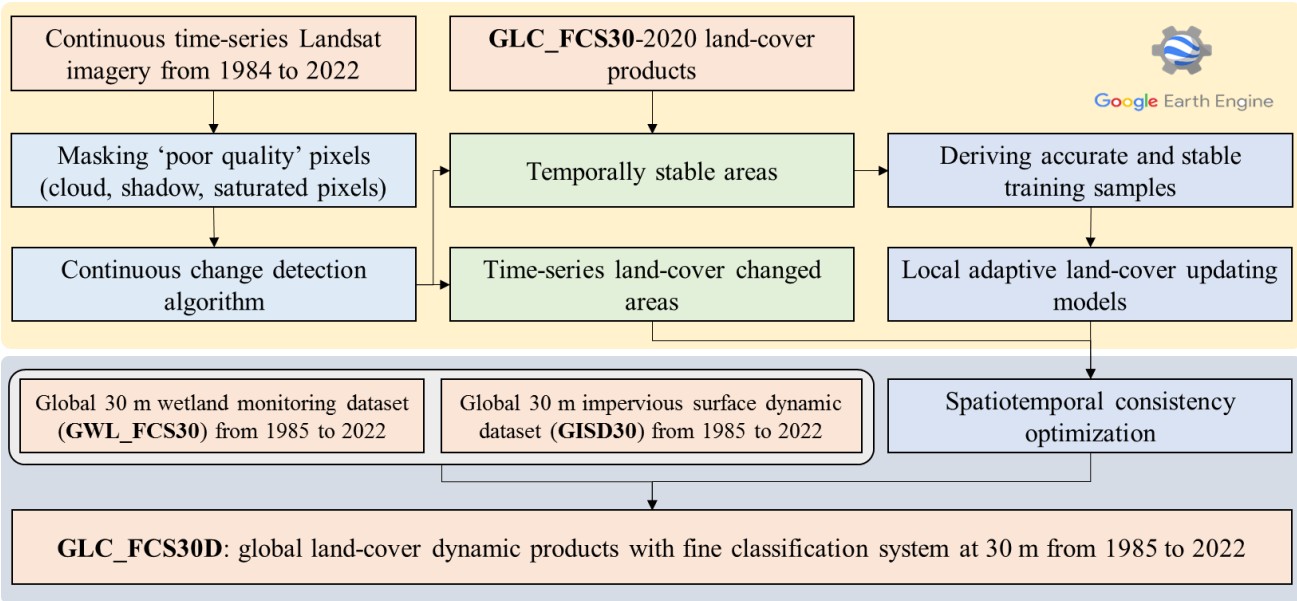

**Figure 2**. The flowchart of the proposed method combining the continuous change-detection (CCD) algorithm and a local adaptive updating algorithm.

## 3.1 The fine classification system used in the GLC_FCS30D

Determining the classification system is usually a prerequisite for land-cover mapping and monitoring. In this study, as we used the GLC_FCS30-2020 as the baseline land-cover product, and further overlaid the GWL_FCS30 dataset on the GLC_FCS30D to ensure high accuracy in the wetland areas; thus, the fine classification system used in this study would inherit from that of the GLC_FCS30-2020 and GWL_FCS30. Table 1 lists the details of the fine classification system. It contains 35 fine land-cover types and has obvious advantages over identifying the forest and wetland subcategories.

**Table 1**. The details of the fine classification system in the GLC_FCS30D land-cover dynamic dataset.

| Basic classification system | | Level-1 validation system | | Fine classification system | Id |
|---|---|---|---|---|---|
| Cropland | CRP | Rainfed cropland | RCP | Rainfed cropland | 10 |
| | | | | Herbaceous cover cropland | 11 |
| | | | | Tree or shrub cover cropland | 12 |
| | | Irrigated cropland | ICP | Irrigated cropland | 20 |
| Forest | FST | Evergreen broadleaved forest | EBF | Closed evergreen broadleaved forest | 51 |
| | | | | Open evergreen broadleaved forest | 52 |
| | | Deciduous broadleaved forest | BDF | Closed deciduous broadleaved forest | 61 |
| | | | | Open deciduous broadleaved forest | 62 |
| | | Evergreen needleleaved forest | ENF | Closed evergreen needleleaved forest | 71 |
| | | | | Open evergreen needleleaved forest | 72 |
| | | Deciduous needleleaved forest | DNF | Closed deciduous needleleaved forest | 81 |
| | | | | Open deciduous needleleaved forest | 82 |
| | | Mixed-leaf forest | MFT | Closed mixed-leaf forest | 91 |
| | | | | Open mixed-leaf forest | 92 |
| Shrubland | SHR | Shrubland | SHR | Shrubland | 120 |
| | | | | Evergreen shrubland | 121 |
| | | | | Deciduous shrubland | 122 |
| Grassland | GRS | Grassland | GRS | Grassland | 130 |
| Tundra | TUD | Lichens and mosses | LMS | Lichens and mosses | 140 |
| Wetland | WET | Inland wetland | IWL | Swamp | 181 |
| | | | | Marsh | 182 |
| | | | | Flooded flat | 183 |
| | | | | Saline | 184 |
| | | Coastal wetland | CWL | Mangrove | 185 |
| | | | | Salt marsh | 186 |
| | | | | Tidal flat | 187 |
| Impervious surface | IMP | Impervious surface | IMP | Impervious surface | 190 |
| Bare areas | BAL | Sparse vegetation | SVG | Sparse vegetation | 150 |
| | | | | Sparse shrubland | 152 |
| | | | | Sparse herbaceous cover | 153 |
| | | Bare areas | BAL | Bare areas | 200 |
| | | | | Consolidated bare areas | 201 |
| | | | | Unconsolidated bare areas | 202 |
| Water body | WTR | Water body | WTR | Water body | 210 |
| Permanent snow and ice | PSI | Permanent snow and ice | PSI | Permanent snow and ice | 220 |

## 3.2 Detecting changes using the CCD algorithm and continuous Landsat imagery

In general, land-cover changes can be grouped into three categories including: periodic changes caused by
250 the phenological variability, trend changes driven by natural behavior (such as vegetation growth), and abrupt

changes caused by natural or human disturbances (such as deforestation, urban expansion). Thus, capturing these abrupt changes and simultaneously suppressing the periodic and trend changes are the key to land-cover monitoring. In this study, the CCD algorithm (Zhu and Woodcock (2014b) captured these abrupt changes. The algorithm uses Fourier transformation to fit the time-series observations with the trend term (estimating the trend changes) and harmonic terms (describing the periodic changes) in Eq. (1).

$$\hat{\rho}(i,t) = a_{0,i} + c_{1,i} \times t + \sum_{k=1}^{n} \left( a_{k,i} \times cos\left(\frac{2k\pi}{T}t\right) + b_{k,i} \times sin\left(\frac{2k\pi}{T}t\right) \right) \tag{1}$$

where $\hat{\rho}(i,t)$ represents the predicted value of the $i$th band at the $t$th Julian day, $c_{1,i}$ and $a_{0,i}$ are the regression slope and intercept of the $i$th band, $a_{k,i}$ and $b_{k,i}$ represent the coefficients of the $k$th order harmonic term for the $i$th band, $n$ denotes the number of harmonic terms, and $T$ is the day number of the year (usually defined as 365). As for how to determine value of $n$, Zhu and Woodcock (2014b) explained that higher order harmonic terms have better performance for capturing the periodic variability, but caused overfitting in the time-series model and needed more clear-sky observations to initialize the coefficients of $a_{k,i}$ and $b_{k,i}$. After balancing the advantages and disadvantages of the higher order harmonic terms, we finally chose $n$ as 3, as suggested by other studies (Xian et al., 2022; Xie et al., 2022).

Then, as the CCD is a multivariate change-detection algorithm for capturing the changes of various land-cover types Zhu (2017), five Landsat spectral bands (excluding the blue band for minimizing the effects of the atmosphere and clouds), and three spectral indexes [including: NDVI, NDWI, and NBR as given in Eq. (2)] were combined to detect many kind of changes in the Landsat time-series.

$$NDVI = \frac{\rho_{nir} - \rho_r}{\rho_{nir} + \rho_r}, \quad NDWI = \frac{\rho_{green} - \rho_{swir1}}{\rho_{green} + \rho_{swir1}}, \quad NBR = \frac{\rho_{nir} - \rho_{swir1}}{\rho_{nir} + \rho_{swir1}} \tag{2}$$

where $\rho_{green}$, $\rho_r$, $\rho_{nir}$ and $\rho_{swir1}$ are the green, red, NIR, SWIR1 and SWIR2 spectral bands in the Landsat imagery. Next, to determine the fitted coefficients of the kth order harmonic term in Eq. (1), the Least Absolute Shrinkage and Selection Operator (LASSO) regression algorithm was applied, which demonstrated better performance than the traditional Ordinary Least Squares method in reducing the overfitting problem and dealing with unevenly distributed and sparse Landsat observations (Zhu and Woodcock (2014b).

Next, the CCD was also a multi-parameter change detection model and demonstrated to be sensitive to the parameter settings (Xiao et al., 2023; Zhu and Woodcock, 2014b). The CCDC algorithm on the Google Earth Engine platform (ee.Algorithms.TemporalSegmentation.Ccdc) contained three key adjustable parameters: minObservations, chiSquareProbability and minNumOfYearScaler. Zhu et al. (2019) analyzed the relationships between the omission error and commission error of land-cover changes with the variability of three parameters in the United States, and found their values affected the change detection accuracy. In this study, we also investigated the sensitivity between parameter settings with the change detection accuracies in Figure S1 (seen the Supplement material) using the time-series points from LCMAP_Val and LUCAS datasets after partly sampling. Notably, the sensitivity analysis was implemented in two large-areas for ensure the feasibility of optimal parameters, that is, which will be suitable for other areas in land-cover change detection. The results also showed the CCD is a parameter-sensitive algorithm and the optimal parameter values were 5, 0.95 and 2-year for minObservations, chiSquareProbability and minNumOfYearScaler.

After modeling the time-series observations using the CCD algorithm, we can analyze the land-cover changes from the differences between actual observations and predicted values in the time-series fitting models. Figure 3 shows three typical scenarios in which land-cover dynamics were modeled by the CCD algorithm. Specifically, Figure 3a illustrates that there was no abrupt break in the whole period and thus only the single time-series model was built, and the pixel was usually labeled as temporally stable. Figure 3b indicates that the

pixel underwent an abrupt change and the time-series observations were split into two segments. The time point of the abrupt change occurred around 1996. Figure 3c gives a complicated time-series disturbance example, in which multiple abrupt changes were detected and the time-series observations were split into four segments. The time-series models for segments 1, 2, and 4 showed obvious trend changes.

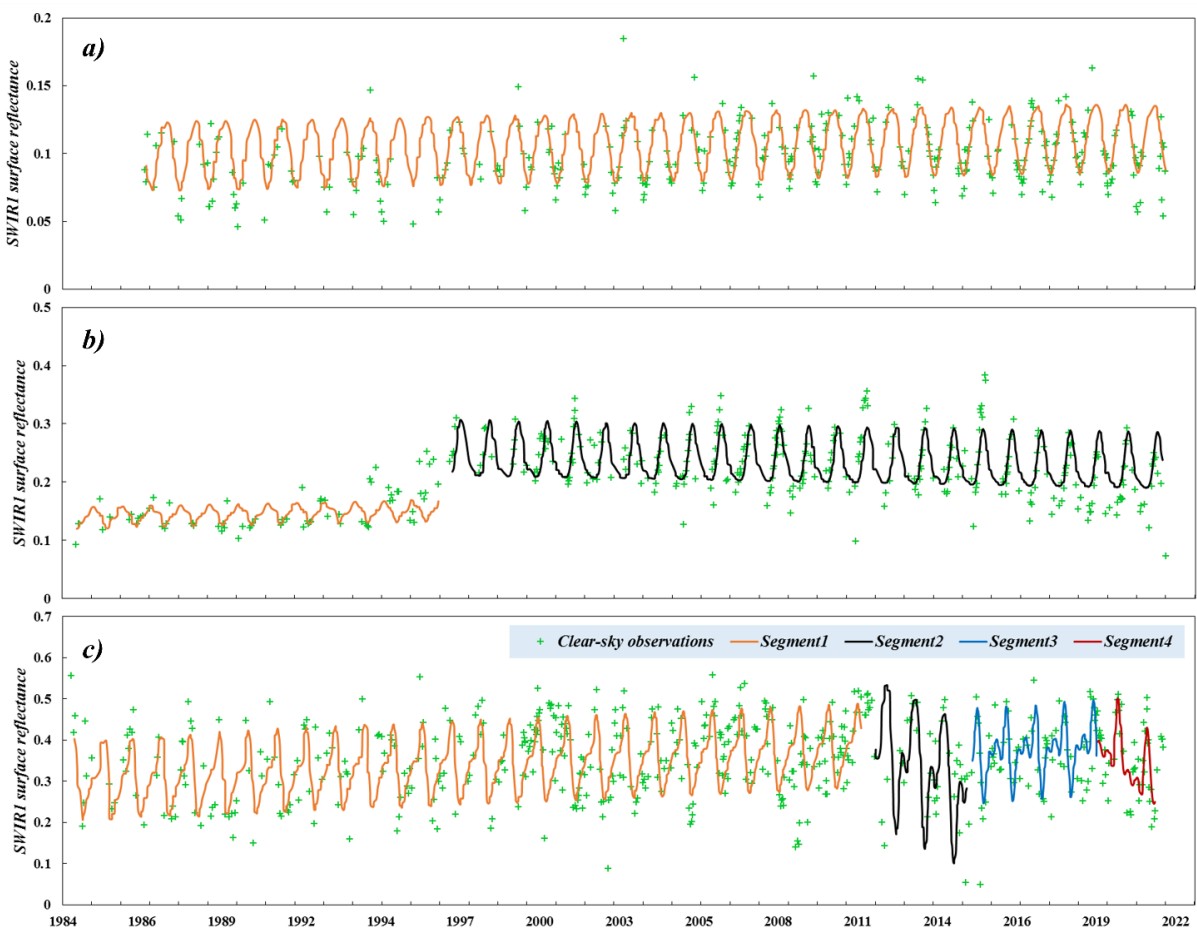

**Figure 3**. Three typical land-cover changes using the continuous change detection (CCD) algorithm and continuous Landsat observations; a) time-series stable land-cover condition; b) one single abrupt change; c) multiple abrupt changes.

### 3.3 Updating changed areas using local adaptive classifications

        Using the CCD algorithm and continuous Landsat imagery, we identified the temporally stable pixels and

the time points of abrupt changes for the land-cover change pixels. Accurately determining land-cover labels of the changed pixels (or understanding the change process 'from-to') is another key procedure for time-series land-cover monitoring. To achieve this goal, we derived spatiotemporally stable training samples (see Section

3.3.1), then updated the changed pixels using multitemporal classifications, and finally minimized the cumulative error caused by independent classifications.

**3.3.1 Deriving spatiotemporally stable training samples**

Numerous studies demonstrated that the accurateness of the training samples plays a critical role in accurate mapping (Foody and Arora, 2010; Zhang et al., 2020). Visual interpretation can ensure high-confidence samples at the expense of a large quantity of manual participations, so it was not suitable for collecting large-area training samples. An alternative option involved generating training samples by refining existing land-cover products

through a series of improvement measures (Zhang et al., 2021b; Zhang et al., 2023). Inspired by the latter option, we combined the GLC_FCS30-2020 prior dataset and the change-detection mask (derived from the CCD algorithm described in Section 3.2) to obtain the spatiotemporally stable training samples. Specifically, temporally stable areas are known to have higher mapping accuracy (Yang and Huang, 2021; Zhang and Roy, 2017; Zhang et al., 2023); thus, we first used the aforementioned CCD mask to retain these temporally stable

areas during 1985-2022, and then overlap them into the GLC_FCS30-2020 maps to determine their land-cover labels. Next, Radoux et al. (2014) emphasized that land-cover transition areas usually were subject to more serious misclassification problems and that the homogeneous land-cover pixels had a higher probability of achieving acceptable accuracy. Therefore, we used the morphological erosion filter of 3 pixels × 3 pixels to refine these temporally stable areas into spatiotemporally homogeneous areas.

Benefiting from the temporally stable checking during 1985-2022, spatial homogeneity analyzing, and the overall accuracy of 82.5% in GLC_FCS30-2020 products, these spatiotemporally stable areas are retained to generate the training samples. It should be noted that the spatiotemporally stable areas are not guaranteed to be completely accurate, that is, a small number of derived training samples might be mis-labeled. Fortunately, previous studies in large-area land-cover mapping demonstrated that the random forest classification models

(adopted by this study in Section 3.3.2) is highly robust to the erroneous training samples (Gong et al., 2019b; Mellor et al., 2015; Zhang et al., 2021b). For example, Gong et al. (2019b) found that the overall accuracy kept relatively stable when the proportion of erroneous training samples is controlled within 20%. Thus, the used spatiotemporally stable areas can be supported to derive confident training samples and further ensure the quality of land-cover dynamic monitoring.

Numerous studies have highlighted the importance of training sample balance and distribution, as they significantly influence the mapping performance (Foody, 2009; Jin et al., 2014; Millard and Richardson, 2015). First, in term of the sample distribution, there are two options for training sample distribution including areal-proportional or equal allocation, and former was demonstrate to achieve higher accuracy than the latter option in land-cover mapping especially in complicated land-cover conditions (Jin et al., 2014). However, when using

the areal-proportional sampling strategy, the rare land-cover types usually had small sample sizes and would be sacrificed because the aim of land-cover mapping was to achieve a global optimum rather than a local optimum. Thus, the maximum and minimum sample size for abundant and rare land-cover types were suggested as 8000 and 600, came from the study in Zhu et al. (2016), for avoiding the extremes of sample sizes. Next, the GLC_FCS30-2020 products were split into 961 5° × 5° geographical tiles, and we used the areal-proportional

sampling strategy and two sample balancing thresholds to allocate the training samples from the spatiotemporally stable areas in each 5° × 5° geographical tile. Last, the impervious surface and wetland samples were excluded because both have been independently developed as the thematic datasets in Section 2.3 and 2.4.

### 3.3.2 Updating changed areas using local adaptive classifications

Before building the local adaptive classification models, we must extract useful spectral features from the time-series Landsat observations. In this study, we used multitemporal phenological, texture, and topographical features. Specifically, the multitemporal phenological features were extracted by using the percentile-compositing method, which has fewer constraints than other compositing algorithms (such as the seasonal-based compositing method) while achieving similar mapping accuracy (Azzari and Lobell, 2017). The time-series Landsat spectral bands (five optical bands after excluding atmospherically sensitive blue band) and corresponding spectral indexes [NDVI, NDWI, and NBR in Eq. (2)] were composited into five percentiles (10th, 25th, 50th, 75th, and 90th). Next, in terms of the texture features, our previous study explained that the texture features had positive contribution on land-cover mapping (Zhang et al. (2021b), so the gray-level co-occurrence matrix method was used for the 50th-percentile–composited NIR band to extract the homogeneity, entropy, dissimilarity, variance, contrast, and correlation. Last, since the land-cover distribution was usually related to the topographical environment, for example, croplands and water bodies are mainly distributed in flat areas, three topographical variables (elevation, slope, and aspect), calculated from a global 30 m DEM dataset (named as: ASTER_GDEM) (Tachikawa et al., 2011), were also imported. In addition, due to the limited storage capacity and satellite–ground data-transmission capacity of early satellites, the density of Landsat imagery is sparse before 2000 (only Landsat 5 single-satellite acquired data) (Roy et al., 2014b). We choose the coarse temporal cycle of 5-years for ensuring the mapping accuracy before 2000, that is, the satellite observations from two years before and after was used for the nominal center year. For example, we update the land-cover maps in 1995 using all available imagery from 1993 to 1997. In total, there were 49 multisource features, including 40 phenological spectra features, 6 texture features, and 3 topographical variables.

There are two options for global land-cover mapping and updating: global modeling and local adaptive modeling, and our previous studies have explained that local adaptive modeling yields superior results compared to global modeling. This is primarily due to the former's capability to take regional characteristics into account more effectively, leading to increased sensitivity in training samples and higher accuracy in land-cover classification (Zhang et al., 2021b; Zhang et al., 2023; Zhang et al., 2022). Thus, we first inherited the regional gridding style in the GLC_FCS30 (Zhang et al., 2021b), namely, the global land was divided into 961 5° × 5° geographical tiles. Afterward, the local classification models were independently built for updating the land cover in each tile using the corresponding training samples in the neighboring eight surrounding tiles at 3 × 3 window. The adjacent training samples were imported to increase the continuity of the adjacent land-cover maps.

Last, in the selection of the suitable classification algorithm, random forest (RF) classifier has significant advantages, including: accommodating high-dimensional training features, better ability to deal with the overfitting problem, and higher classification accuracy than other widely used classifiers (Belgiu and Drǎguţ, 2016; Gislason et al., 2006). Meanwhile, the RF algorithm was also integrated into the internal function library of the GEE cloud platform as *ee.Classifier.smileRandomForest()*. Thus, the RF algorithm was used to combine the training samples and multisourced features for updating the changed pixels. The RF algorithm allows for adjusting two key parameters (the number of decision tree (Ntree) and predicted variables (Mtry)), and previous studies have quantitatively analyzed the relationships between classification accuracy with the value of these two parameters. Both theoretical and experimental results indicated that the selection of Mtry and Ntree had little influence on the classification accuracy (Belgiu and Drǎguţ, 2016; Du et al., 2015). Thus, the default

recommended values of 500 for Ntree and the square of the total number of input features for Mtry were used based on previous studies (Belgiu and Drăguţ, 2016; Zhang et al., 2019).

### 3.3.3 Temporal-consistency optimization

To ensure the rationality and consistency of land-cover changes for long time-series, the CCD algorithm was applied to capture the time points of land-cover changes, and then the changed pixels were updated using the local adaptive classifications. In this study, despite our best efforts, it was difficult to completely eliminate classification errors, particularly when dealing with changes over time. To address this issue and enhance accuracy in areas with temporal variations, we employed the temporal consistency optimization method described in Eq. (3). This approach incorporates both temporal and spatial neighboring information to assess homogeneity, thereby reducing potential misclassifications in time-series changed areas.

$$P_{x,y,t} = \frac{1}{N}\left[\sum_{x\prime=x-1}^{x\prime=x+1}\sum_{y\prime=y-1}^{y\prime=y+1}\sum_{t\prime=t-1}^{t\prime=t+1}I\big(L_{x\prime,y\prime,t\prime} = L_{x,y,t}\big)\right] \tag{3}$$

Where $P_{x,y,t}$ is the homogeneity probability of the pixel in spatial location $(x, y)$ and time point $t$; usually, the higher the value of $P_{x,y,t}$, the less the classification error effect. $L_{x,y,t}$ and $L_{x\prime,y\prime,t\prime}$ are the land-cover labels of the central pixel and the corresponding spatiotemporal neighboring pixels with a local window of $3 \times 3 \times 3$, and the $I()$ denotes the indicator function for the equation of the status between two pixels. Namely, if $L_{x\prime,y\prime,t\prime}$ was equal to the $L_{x,y,t}$, then the value of indicator function was 1, otherwise it was equal to 0 (Kenny, 2003). In this study, the homogeneity probability was calculated for each changed pixel, and used the threshold of 0.5 (as suggested by and used in the studies of (Li et al., 2015; Zhang et al., 2022) to judge the rationality of land-cover changes. Namely, if the $P_{x,y,t}$ was less than the threshold, the $L_{x,y,t}$ would be modified according to the spatiotemporal pixels.

## 3.4 Accuracy assessment

The validation process for the GLC_FCS30D dataset follows the recommended guidelines proposed by Pontus Olofsson (2014). These guidelines encompass two key components: area estimation (nonsite-specific accuracy) and accuracy assessment (site-specific accuracy). The site-specific accuracy assessment mainly focuses on estimating the confusion matrix and calculating some accuracy metrics including overall accuracy (O.A.), producer's accuracy (P.A.), user's accuracy (U.A.) and the corresponding standard errors using a poststratified estimator (Pontus Olofsson, 2014).

$$P.A._{\cdot k} = \frac{p_{kk}}{\sum p_{k\cdot}}, U.A._{\cdot k} = \frac{p_{kk}}{\sum p_{\cdot k}}, O.A. = \sum_{k=1}^{m} p_{kk} \tag{4}$$

Where $p_{kk}$ was the proportion of the area mapped as class $k$ that had reference class $k$, $\sum p_{k\cdot}$ and $\sum p_{\cdot k}$ were the proportion of the area mapped as class $k$ and the proportion of the reference area as class $k$, and the $m$ denoted the number of land-cover types. Afterwards, because there is currently no global long-time series validation dataset, we used 84526 global validation points to assess the accuracy metrics of the GLC_FCS30D dataset in 2020 and used two third-party datasets to analyze the time-series accuracy variations. The GLC_FCS30D adopts a fine classification system containing 35 subcategories, for which we applied an analysis protocol into the basic classification system and the LCCS level-1 validation system, whose details were explained in the Table 1 and contained 10 major land-cover types and 17 fine land-cover types, respectively. Lastly, to quality the performance of land-cover changed pixels, we followed the proposal of Stehman et al. (2021) in assessing the LCMAP annual land-cover products 1985-2017, that is, the validation pixels were

grouped into "changed" and "unchanged" categories and the corresponding confusion matrix were calculated. Meanwhile, to minimize the imbalance in the sample size of "change" and "no-change" samples, the metrics of F1 score was supplemented as:

$$F1 = \frac{P.A. \times U.A.}{P.A. + U.A.} \times 2 \times 100\% \qquad (5)$$

## 4. Results and discussion

### 4.1 Overview of GLC_FCS30D maps and their changes

Figure 4 provides an overview of the GLC_FCS30D dataset in 2022 (the overview of GLC_FCS30D in 1985 is also given in the Figure S3); overall, it aligns with the real-world land-cover patterns on a global scale. Forest, cropland, barren land, and grassland are the dominant land-cover types, and each of them is distributed in the corresponding ecology subregions. For example, needle-leaved forest is mainly concentrated in the high-latitude cold regions while broad-leaved forests are mainly distributed in tropical regions; permanent ice and snow is mainly located in Greenland and high-altitude montains. The GLC_FCS30D has significant advantages over other global land-cover datasets in terms of land-cover type diversity; it contains 35 discrete land-cover types, among which forest and wetland are subdivided into 10 and 7 land-cover subcategories, respectively.

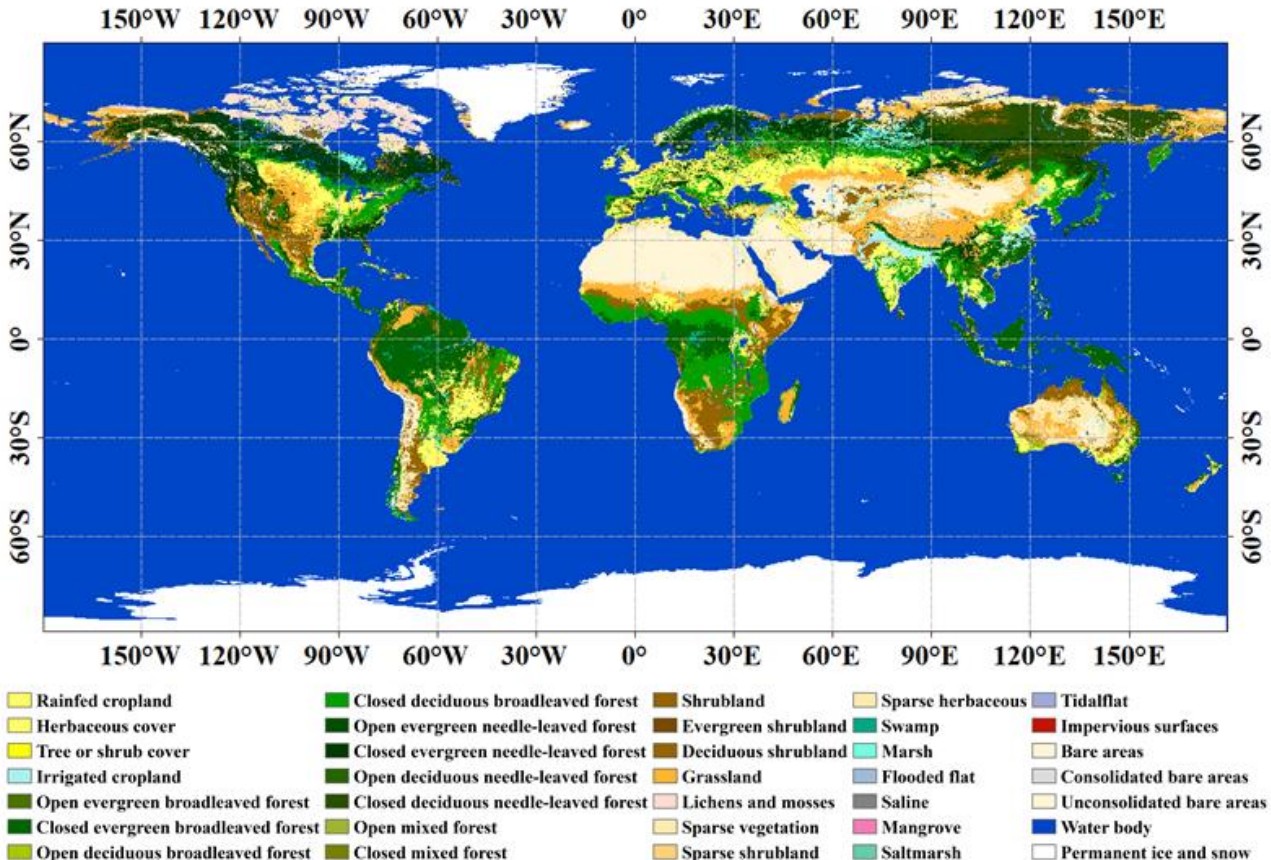

**Figure 4**. Overview of the GLC_FCS30D in 2022 with a color-coded legend derived from the European Space Agency (ESA) Climate Change Initiative land-cover dataset (Defourny et al., 2018).

Figure 5a illustrates the spatial distribution of land-cover change intensity (measuring the proportions of changed pixels in the 0.05° grid) in the GLC_FCS30D from 1985 to 2022 after upscaling to a resolution of 0.05°. Obviously, global land-cover has experienced significant changes over the past 37 years, mainly in the

following three typical areas: 1) tropical rainforest peripheral areas in South America and Southeast Asia, in which deforestation is the dominant cause; 2) wetland and water-body intermingling areas, such as North America and northern Asia, in which water bodies and wetland were transformed into one another due to different annual water levels. In the GLC_FCS30D the water body land-cover type represents permanent water during the year (it may be wetland in other years). 3) The semi-arid areas in Australia, Central Asia, and western Africa, where land cover (such as sparse vegetation or bare land) is directly affected by precipitation and temperature. For example, if there is sufficient precipitation in the year, the sparse vegetation and some bare land would be covered by grass in semi-arid areas. Similarly, the work of Winkler et al. (2021) revealed that these semi-arid areas experienced serious and frequent land-cover changes. Figure 5b quantitatively counts the changed areas of 10 major land-cover types from 1985 to 2022. Forest and cropland variations dominated global land-cover change. The net loss of forests over the past 37 years reached approximately 2.5 million $km^2$, and the decline is steady over time. Conversely, cropland showed a stable increase and the net gain in cropland area is approximately 1.3 million $km^2$. Shrubland, wetland, and impervious surface had increased areas of 0.45 million $km^2$, 0.40 million $km^2$, and 0.37 million $km^2$, respectively. The increased shrubland resulted from the recovery of deforested land, and the wetland gains are due to increases in seasonal water bodies. The work of Pekel et al. (2016) emphasized that the global seasonal water bodies, labeled as inland wetland in the GLC_FCS30D, showed an overall increase.

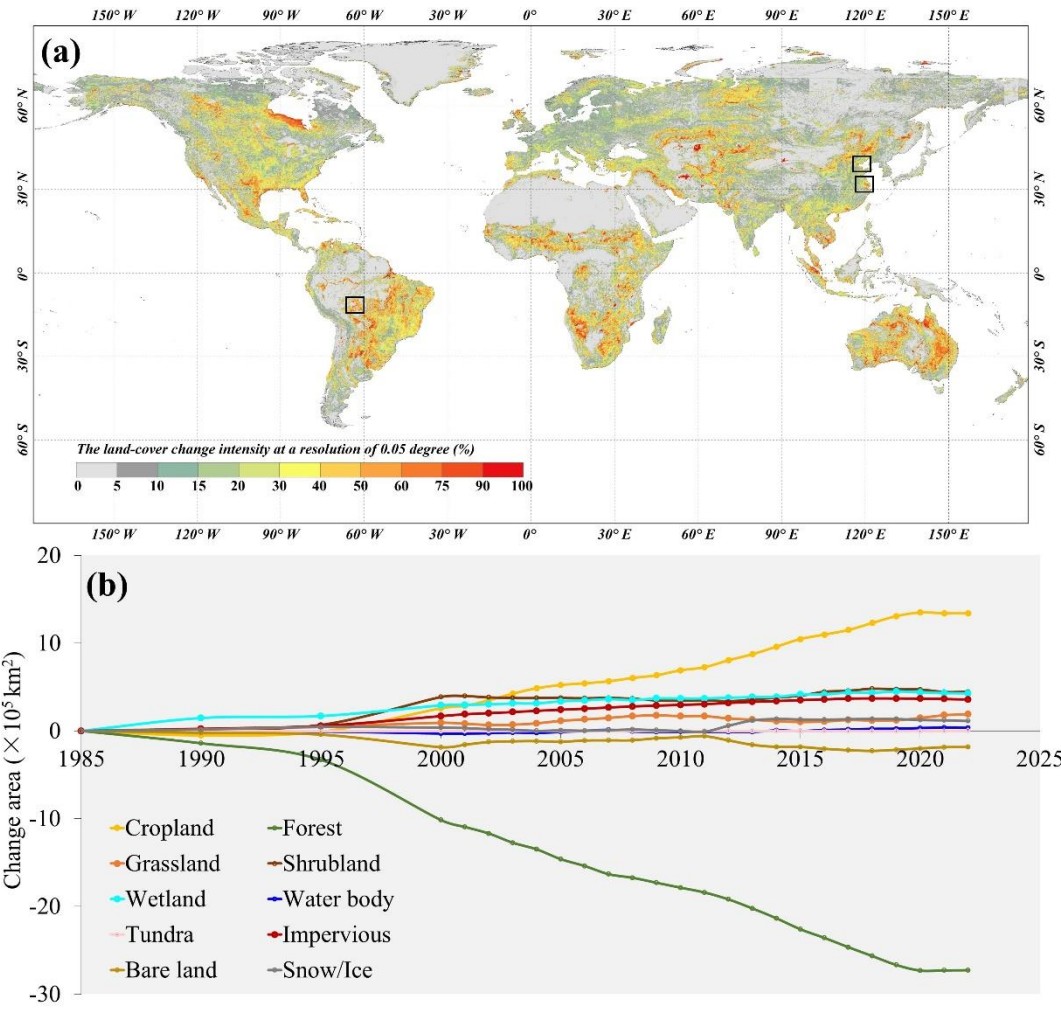

**Figure 5**. (a) The spatial distribution of global land-cover change intensity from 1985 to 2022 after aggregating to a resolution of 0.05°. (b) The net areas of 10 major land-cover types in GLC_FCS30D from 1985 to 2022.

Figure 6 further analyzes the net area variations of 10 major land-cover types on six continents. The six continents exhibit various land-cover change characteristics, for example, steady forest loss and cropland gain dominate land-cover change in South America, while the net area variations of most land-cover types fluctuate in Australia. North America experiences obvious deforestation, and the forest loss area reaches approximately $4.5 \times 10^5$ km$^2$. In contrast, shrubland, grassland, and impervious surface land-cover types show an overall increasing trend, with increases of $1.4 \times 10^5$ km$^2$, $0.8 \times 10^5$ km$^2$, and $0.72 \times 10^5$ km$^2$, respectively. Similarly, Xian et al. (2022) reported that forest losses and shrubland, grassland, and impervious surface gains are the dominant characteristics of the CONUS from 1985 to 2017. In Europe, the forest area continues to decrease, and the cropland area first decreased and then increased because of the collapse of the Soviet Union in 1990s. Abandoned croplands were transformed into pasture (which also belongs to the cropland land-cover type in the GLC_FCS30D). In Asia, the increase in impervious surface is the most significant across the six continents with a net increase of $1.9 \times 10^5$ km$^2$; wetland also shows a large increase of $1.1 \times 10^5$ km$^2$. The increased wetland coverage comes from the increase in seasonal water bodies. South America and Africa experience similar land-cover change characteristics, with the most intense deforestation rates and the most significant increases in cropland. According to our statistics, the forest loss on these two continents amounts to $16.9 \times 10^5$ km$^2$ and the corresponding increase of cropland is approximately $11.1 \times 10^5$ km$^2$. Last, because Oceania is more sensitive to climate change, especially in terms of precipitation, the fluctuations of shrubland, grassland, and bare land are evident because the conversion relationship between the three land-cover types is related to annual precipitation.

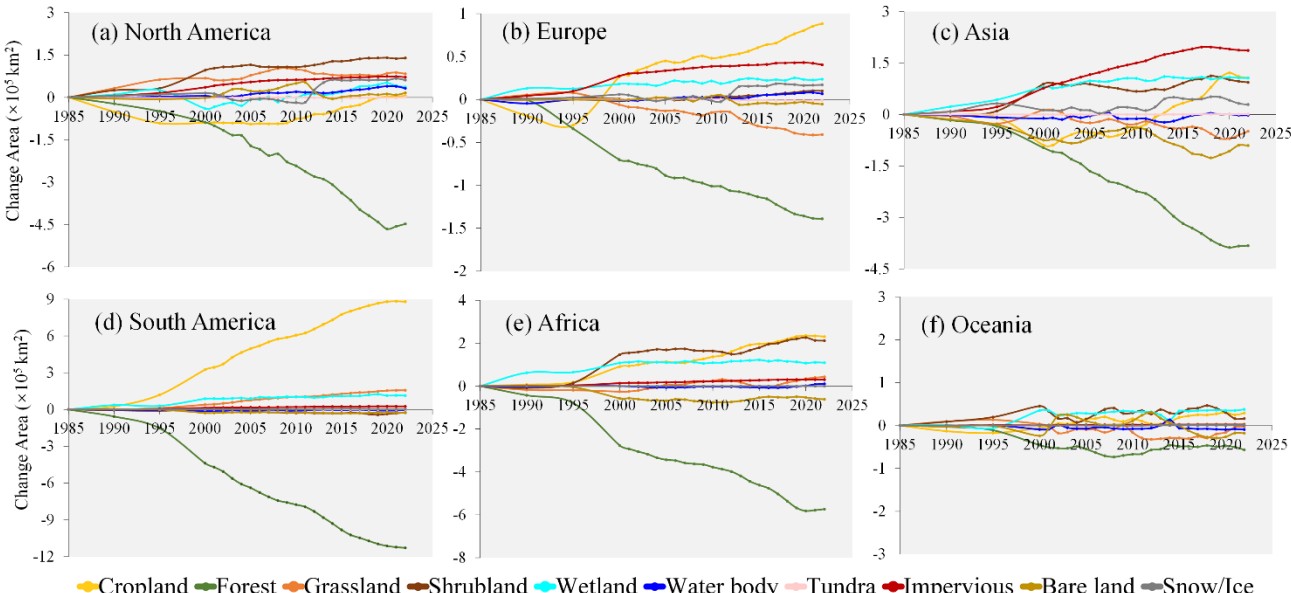

**Figure 6**. The net area variations of 10 major land-cover types on six continents from 1985 to 2022.

Figure 7 displays the land-cover transformation relationships from 1985 to 2022 in the GLC_FCS30D dataset using Sankey diagrams. Global cropland and forest have obvious area changes and area proportions have changed from 12.08% and 38.26% in 1985 to 12.86% and 36.48% in 2022. Shrubland changed from 8.70% in 1985 to 9.03% in 2022. We mainly focus on forest, cropland, shrubland and impervious surface changes, which dominate the land-cover changes in Figure 5. There are three main causes of forest loss over the past 37 years:

1) 37.58% of deforested land was converted to cropland, which was more significant in tropical rainforest areas (Figure 8a); 2) 26.92% of the lost forest was regrown as shrubland, which is more common in mountainous areas affected by wildfires; and 3) 13.49% of deforested land was converted to grassland. Cropland is converted to forest, grassland, and impervious surface. A total of 26.29% of lost cropland is converted to grassland due to abandonment, 25.88% of lost cropland is covered by forests, and 21.01% of lost cropland resulted from urbanization. Lastly, regarding impervious surface, our primary focus was on identifying the sources contributing to its expansion. Our findings indicate that approximately 36.24% of the impervious surface increase can be attributed to the conversion of cropland, while 13.49% of the increase is a result of deforestation.

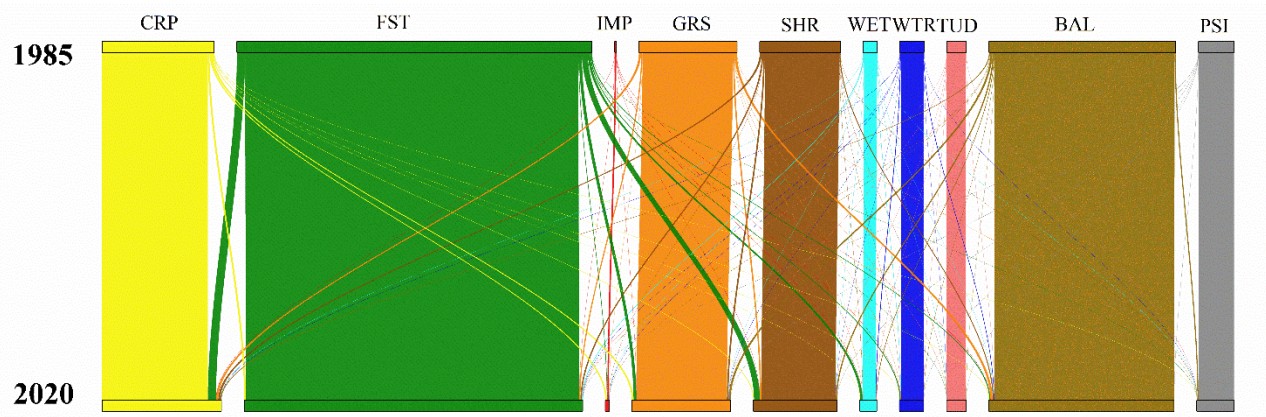

**Figure 7**. Sankey diagrams of the global land-cover changes during 1985-2022 in the GLC_FCS30D dataset.

To visually understand the land-cover change process captured by the GLC_FCS30D dataset over past 37 years, Figure 8 displays three typical enlargements (spatial location illustrated as two black rectangles in Figure 5a) of the Amazon rainforest (which experienced significant deforestation) and China's Yangtze River Delta (which underwent rapid urbanization) and Yellow River Delta (evident land-cover changes over coastal regions). These three typical areas experienced drastic land-cover changes and the GLC_FCS30D accurately captures the spatiotemporal changes. Specifically, the deforestation in South America is widely recognized, and the GLC_FCS30D clearly reflects this trend. Namely, the early deforestation showed a grid distribution, and then each grid gradually extended outward and finally connected into patches. The GLC_FCS30D also shows that deforestation has not stopped in the region in terms of the rate of forest loss, and these findings are in line with the results of earlier researches (Harris et al., 2021; Potapov et al., 2022). In the Yangtze River Delta, GLC_FCS30D depicts that the dominant land-cover change over the enlargement is urbanization, and a large quantity of irrigated cropland has been converted to impervious surfaces. Meanwhile, urban expansion was significantly faster before 2010 than after 2010 with the GLC_FCS30D. Lastly, the Yellow River Delta, as one of the typical coastal region, was selected to understand the GLC_FCS30D for capturing these coastal land-cover changes. Obviously, the land-cover changes in the GLC_FCS30D can be concluded into three aspects: 1) a large amount of flooded flats and flat flats were reclaimed as the aquaculture ponds, especially after 2000; 2) the mouth of the Yellow River turned from south to north (black rectangle), that is, there were large land-cover changes between tidal/flooded flats, water bodies and salt marshes; 3) a lot of impervious surfaces encroached the coastal water-bodies and flats. In short, if we combine real time-series remote-sensing observation data, the GLC_FCS30D effectively captures the spatiotemporal changes of the land surface.

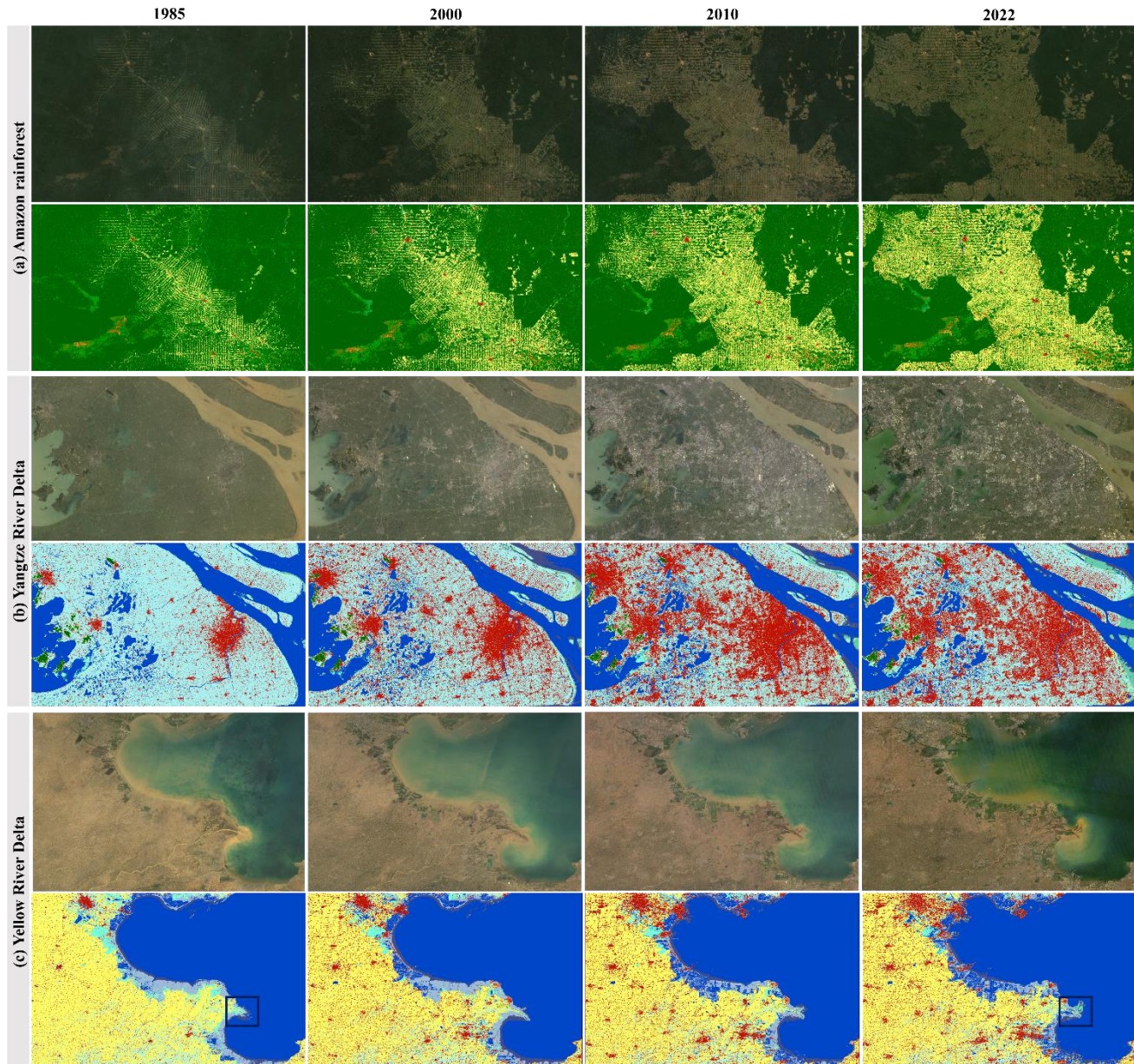

**Figure 8**. Three typical enlargements of land-cover changes in the GLC_FCS30D from 1985 to 2022 in (a) the Amazon rainforest, (b) the Yangtze River Delta in China, and (c) the Yellow River Delta in China. The color-coded legend is like the global map in Figure 4. In each case, the natural-color imagery from 1985 to 2022 is a composite taken from Landsat imagery.

## 4.2 Accuracy assessment of the GLC_FCS30D in 2020

Table 2 provides the error matrix and accuracy metrics for the GLC_FCS30D dataset in the basic classification system containing 10 major land-cover types. The novel GLC_FCS30D dataset attained an O.A. of 80.88% ($\pm$0.27%). The cropland, forest, impervious surface, water body, as well as permanent snow and ice perform better in terms of the P.A. and U.A. than the remaining land-cover types, with corresponding accuracies exceeding 85%. The impervious surface and wetland datasets are independently generated and then overlaid on the GLC_FCS30D, helping these complicated land-cover types achieve high accuracy metrics. Conversely, grassland, shrubland, and tundra have lower accuracies; for example, grassland had the lowest P.A. of 54.41%

and shrubland had the lowest U.A. of 57.63%. The two reasons that they performed poorly were as follows: 1) these land-cover types usually reflected heterogeneous and varied spectral and spatial characteristics, such as the grassland shared similar spectra with cropland and sparse shrubland in the growing season and mimicked bare-land features in harvest season; 2) all of them were distributed in climate-transition areas with complicated climate variations and landscapes.

**Table 2.** Error matrix of the GLC_FCS30D dataset in 2020 based on the basic classification system. The reported Producer's Accuracy (P.A.) and User's Accuracy (U.A.) come with their corresponding standard errors (SE) shown in parentheses.

| Reference | Map CRP | FST | GRS | SHR | WET | WTR | TUD | IMP | BAL | PSI | Total | P.A.(SE) |
|---|---|---|---|---|---|---|---|---|---|---|---|---|
| | | | | | O.A. = 80.88% (±0.27%) | | | | | | | |
| CRP | 15.442 | 0.792 | 0.679 | 0.388 | 0.086 | 0.027 | 0 | 0.174 | 0.117 | 0 | 17.704 | 87.22(0.54) |
| FST | 0.513 | 28.712 | 0.315 | 0.811 | 0.371 | 0.021 | 0.008 | 0.063 | 0.113 | 0.002 | 30.93 | **92.83(0.31)** |
| GRS | 1.035 | 1.166 | 5.906 | 1.181 | 0.231 | 0.011 | 0.084 | 0.051 | 1.181 | 0.01 | 10.855 | **54.41(1.02)** |
| SHR | 0.555 | 1.798 | 0.863 | 5.392 | 0.161 | 0.013 | 0.019 | 0.05 | 0.502 | 0.002 | 9.356 | **57.63(1.09)** |
| WET | 0.068 | 0.465 | 0.156 | 0.157 | 4.047 | 0.347 | 0.031 | 0.021 | 0.222 | 0.001 | 5.516 | 73.37(1.27) |
| WTR | 0.04 | 0.086 | 0.019 | 0.017 | 0.302 | 3.305 | 0.008 | 0.012 | 0.039 | 0.002 | 3.831 | 86.28(1.12) |
| TUD | 0.01 | 0.123 | 0.168 | 0.167 | 0.018 | 0.03 | 2.444 | 0.002 | 0.473 | 0.02 | 3.454 | 70.76(1.65) |
| IMP | 0.084 | 0.058 | 0.024 | 0.04 | 0.001 | 0.006 | 0.002 | 5.043 | 0.024 | 0 | 5.283 | 95.45(0.61) |
| BAL | 0.13 | 0.049 | 0.783 | 0.585 | 0.043 | 0.045 | 0.577 | 0.048 | 9.239 | 0.131 | 11.628 | 79.45(0.8) |
| PSI | 0 | 0.004 | 0.03 | 0.005 | 0 | 0.023 | 0.001 | 0 | 0.03 | 1.351 | 1.443 | 93.63(1.38) |
| Total | 17.877 | 33.251 | 8.943 | 8.743 | 5.259 | 3.828 | 3.176 | 5.464 | 11.94 | 1.52 | | |
| U.A.(SE) | 86.38 (0.55) | 86.35 (0.4) | 66.05 (1.07) | 61.68 (1.11) | 76.96 (1.2) | 86.33 (1.35) | 76.97 (1.6) | 92.29 (0.77) | 77.38 (0.82) | 88.89 (1.72) | | |

Note: The abbreviations correspond to the 10 categories of the basic classification system in Table 1.

Table 3 provides the error matrix of the GLC_FCS30D in 2020 in the LCCS level-1 validation system with 17 land-cover types. The GLC_FCS30D-2020 dataset achieves an O.A. of 73.04% (±0.30%), which is lower than that in the basic classification system, because these similar land-cover subcategories more easily suffer from misclassifications. For example, forest has a P.A. of 92.83% (±0.31%) and the P.A. rapidly decreases to the range of 58.29% (±1.53%) to 82.39% (±0.98%) when split into five fine subcategories. Cropland, forest, and bare land, which are further divided into multiple subcategories, show obvious decreases in accuracy over their subcategories in terms of P.A. and U.A. Taking cropland and forest as examples, approximately 31.7% of irrigated cropland (ICP) is misclassified as rainfed cropland (RCP) and so the U.A. of ICP is only 59.92%. More than 53.8% of mixture forests (MFT) are wrongly labeled as the other four forest subcategories and so the mixture forests have the lowest U.A. of 39.34% (±1.38%). Meanwhile, sparse vegetation has the second lowest U.A. of 50.63% (±1.47%) because of the confusion among sparse vegetation, grassland, and bare land. In the basic classification system (Table 1), sparse vegetation is grouped in with bare land. A previous study in Europe Union proposed grouping it as grassland (Gao et al., 2020). Wetland is further divided into coastal wetland (CWL) and inland wetland (IWL) in Table 3, and the CWL has higher U.A. than that of wetland in Table 2, primarily attributed to its significantly more accurate classification in the CWL (Zhang et al., 2023).

**Table 3.** Error matrix of the GLC_FCS30D dataset in 2020 based on the LCCS level-1 validation system. The reported Producer's Accuracy (P.A.) and User's Accuracy (U.A.) come with their corresponding standard errors (SE) shown in parentheses.

| Reference | RCP | ICP | EBF | DBF | ENF | DNF | MFT | SHR | GRS | LMS | SVG | IWL | CWL | IMP | BAL | WTR | PSI | Total | P.A. (SE) |
|---|---|---|---|---|---|---|---|---|---|---|---|---|---|---|---|---|---|---|---|
| RCP | 12.225 | **1.023** | 0.239 | 0.358 | 0.102 | 0.016 | 0.009 | 0.382 | 0.66 | 0 | 0.078 | 0.056 | 0.005 | 0.124 | 0.028 | 0.001 | 0 | 15.332 | 79.7(0.7) |
| ICP | 0.397 | 1.932 | 0.026 | 0.016 | 0.005 | 0 | 0 | 0.01 | 0.025 | 0 | 0.012 | 0.029 | 0.005 | 0.052 | 0 | 0.018 | 0 | 2.527 | 76.45(1.81) |
| EBF | 0.2 | 0.048 | 9.091 | 1.098 | 0.262 | 0.103 | **0.151** | 0.371 | 0.084 | 0 | 0.012 | 0.136 | 0.028 | 0.029 | 0.001 | 0.004 | 0 | 11.514 | 78.96(0.82) |
| DBF | 0.187 | 0.016 | 0.632 | 6.838 | 0.537 | 0.294 | **0.396** | 0.235 | 0.144 | 0.002 | 0.019 | 0.077 | 0.002 | 0.025 | 0.005 | 0.004 | 0.002 | 9.054 | 75.53(0.97) |
| ENF | 0.046 | 0.004 | 0.174 | 0.316 | 5.681 | 0.328 | **0.439** | 0.128 | 0.034 | 0.006 | 0.043 | 0.094 | 0 | 0.008 | 0.01 | 0.01 | 0 | 6.895 | **82.39(0.98)** |
| DNF | 0.008 | 0 | 0.002 | 0.13 | 0.245 | 1.854 | **0.073** | 0.071 | 0.053 | 0 | 0.011 | 0.025 | 0 | 0.001 | 0.007 | 0.002 | 0 | 2.414 | 76.79(1.85) |
| MFT | 0.004 | 0 | **0.019** | **0.176** | **0.234** | **0.013** | **0.828** | 0.014 | 0.004 | 0 | 0 | 0.010 | 0.05 | 0 | 0.001 | 0 | 0 | 1.308 | **58.29(1.53)** |
| SHR | 0.518 | 0.042 | 0.299 | 0.9 | 0.328 | 0.131 | 0.034 | 5.44 | 0.871 | 0.019 | 0.441 | 0.157 | 0.005 | 0.05 | 0.065 | 0.013 | 0.002 | 9.438 | 57.63(1.09) |
| GRS | 0.947 | 0.097 | 0.167 | 0.582 | 0.209 | 0.154 | 0.024 | 1.191 | 5.958 | 0.085 | 0.974 | 0.229 | 0.006 | 0.052 | 0.217 | 0.008 | 0.01 | 10.95 | 54.41(1.02) |
| LMS | 0.006 | 0.004 | 0.001 | 0.022 | 0.044 | 0.053 | 0.001 | 0.168 | 0.169 | 2.465 | 0.379 | 0.02 | 0.001 | 0.002 | 0.098 | 0.026 | 0.02 | 3.484 | 70.76(1.65) |
| SVG | 0.064 | 0.01 | 0.008 | 0.006 | 0.007 | 0.01 | 0.001 | 0.397 | 0.462 | 0.025 | 2.71 | 0.012 | 0 | 0.013 | 0.643 | 0.002 | 0.024 | 4.399 | 61.6(1.57) |
| IWL | 0.01 | 0.002 | 0.044 | 0.029 | 0.103 | 0.022 | 0.002 | 0.048 | 0.017 | 0.008 | 0.042 | 2.673 | 0.024 | 0.001 | 0.012 | 0.224 | 0 | 3.263 | **81.91(1.45)** |
| CWL | 0.004 | 0.002 | 0.008 | 0.002 | 0.004 | 0.002 | 0.004 | 0.008 | 0.006 | 0 | 0.008 | 0.188 | 1.476 | 0.007 | 0.007 | 0.059 | 0 | 1.783 | **82.77(1.92)** |
| IMP | 0.074 | 0.011 | 0.008 | 0.008 | 0.037 | 0.002 | 0 | 0.041 | 0.024 | 0.002 | 0.014 | 0.004 | 0 | 5.087 | 0.01 | 0.004 | 0 | 5.329 | 95.45(0.61) |
| BAL | 0.048 | 0.01 | 0.002 | 0.004 | 0.002 | 0.001 | 0 | 0.193 | 0.328 | 0.557 | 0.582 | 0.043 | 0.002 | 0.035 | 5.384 | 0.029 | 0.108 | 7.33 | 73.45(1.11) |
| WTR | 0.014 | 0.024 | 0.014 | 0.014 | 0.019 | 0.008 | 0.006 | 0.011 | 0.016 | 0.007 | 0.011 | 0.168 | 0.114 | 0.011 | 0.019 | 3.054 | 0.002 | 3.509 | 87.04(1.22) |
| PSI | 0 | 0 | 0 | 0.001 | 0.002 | 0 | 0 | 0.005 | 0.03 | 0.001 | 0.011 | 0 | 0 | 0 | 0.019 | 0.023 | 1.363 | 1.455 | 93.65(1.37) |
| Total | 14.757 | 3.224 | 10.753 | 10.56 | 7.833 | 3.724 | 1.97 | 8.711 | 8.883 | 3.179 | 5.353 | 3.927 | 1.668 | 5.497 | 6.526 | 3.482 | 1.532 | | |
| U.A. (SE) | 82.85 | **59.92** | 84.55 | 64.76 | 72.52 | 49.77 | **39.34** | 62.44 | 67.07 | 77.55 | **50.63** | **68.07** | **88.49** | 92.54 | 82.5 | 87.73 | 88.96 | | |
| | (0.67) | **(1.85)** | (1.75) | (1) | (1.08) | (1.76) | **(1.38)** | (1.11) | (1.07) | (1.59) | **(1.47)** | **(1.6)** | **(1.68)** | (0.76) | (1.01) | (1.19) | (1.72) | | |
| O.A. | **73.04% (±0.30%)** | | | | | | | | | | | | | | | | | | |

Note: The abbreviations correspond to the 17 categories of the LCCS validation system in Table 1.

## 4.3 Accuracy assessment based on two third-party regional validation datasets

### 4.3.1 Time-series accuracy metrics of GLC_FCS30D from LCMAP_Val dataset

Figure 9 displays time-series variations of the overall accuracy of the GLC_FCS30D dataset using the LCMAP_Val annual validation dataset from 1985 to 2018 over the CONUS. The GLC_FCS30D achieves a mean O.A. of 79.50% (±0.50%) and varies from a high value of 80.04% (±0.49%) in 2015 to a low value of 78.91% (±0.51%) in 2000. The overall accuracy of GLC_FCS30D is slightly lower at the early stage, which might be related to the density of Landsat observations. The early Landsat missions had weaker satellite-to-ground transmission and onboard recording capabilities (Roy et al., 2014a), so phenological variability and land-cover changes were more difficult to capture in the early stage.

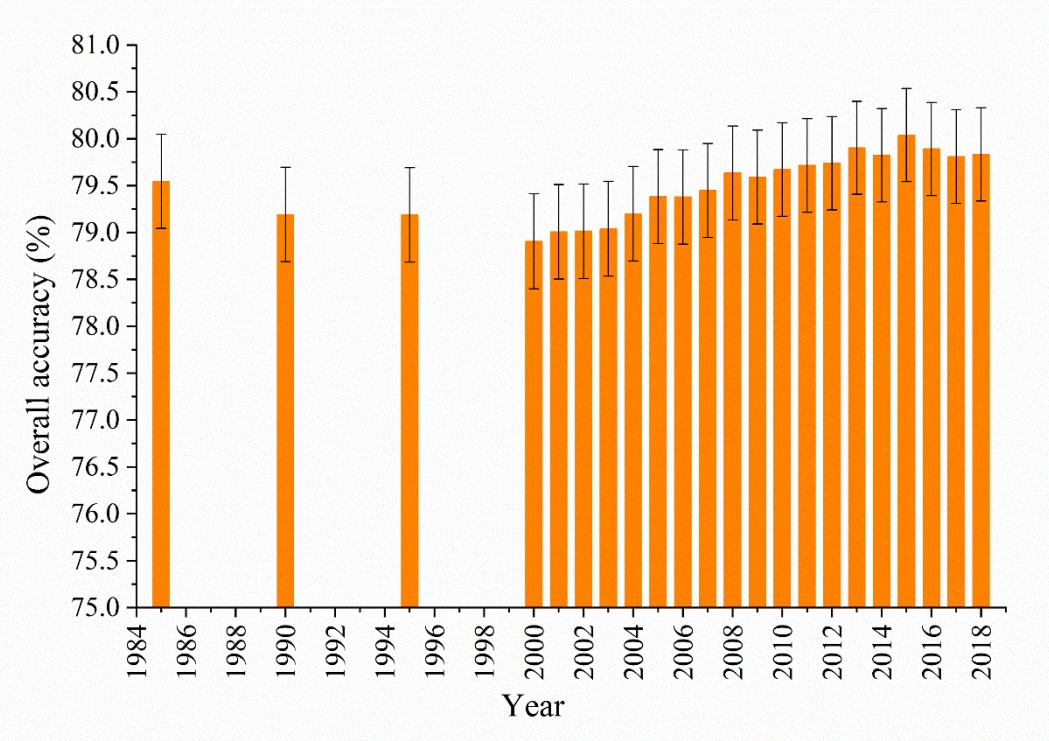

**Figure 9.** The time-series overall accuracy of the GLC_FCS30D dataset using the LCMAP_Val annual reference dataset across the contiguous United States (CONUS) from 1985 to 2018. The error bars on the graph show the uncertainty of each data point.

Figure 10 further illustrates the time-series variations of P.A. and U.A. for the GLC_FCS30D dataset in the CONUS. Visually, the P.A. and U.A. of 10 major land-cover types range from 45% to 100% and 35% to 100%, respectively, and the time-series variations are stable. Among them, the water body land-cover type has the highest accuracy metrics, achieving mean P.A. and U.A. values of 95.31% (±1.14%) and 98.53% (±0.66%), respectively, which benefit from its unique spectral characteristics. Cropland follows with mean P.A. and U.A. values of 93.37% (±0.74%) and 87.70% (±0.94%), respectively. Forest ranks third with a high P.A. of 97.75% (±0.35%) but relatively low U.A. of 82.42% (±0.82%); the unbalanced metrics are because GLC_FCS30D and LCMAP_Val have different definitions for forest. GLC_FCS30D defines the tree cover of the forest as greater than 15% and the threshold setting of LCMAP_Val is 10%, so many shrublands in the GLC_FCS30D are labeled as forest in the LCMAP_Val. Wetland has a U.A. value of 90.47% (±2.05%) but a

P.A. value of 57.07% ( ± 2.75%), which is also caused by a discrepancy in the definition of wetland. GLC_FCS30D identifies seasonal water bodies as wetland while the LCMAP_Val classifies them as water body. Impervious surface has a P.A. lower than 60% mainly because the GLC_FCS30D and LCMAP_Val datasets have different definitions of impervious surface. LCMAP_Val defines buildings and the surrounding green areas as developed, while GLC_FCS30D only includes the artificial buildings (houses, roads, squares, and so on). Bare land and shrubland have the lowest U.A. values of 35.58% ( ± 4.39%) and 47.29% ( ± 1.56%), respectively, mainly because both of them are easily confused with grassland due to the complicated spectral characteristics and coexist in climate-sensitive semi-arid regions (e.g., the Midwestern United States). Xian et al. (2022) emphasized that long-term monitoring of shrubs and grasslands presents significant challenges in the CONUS. Permanent snow and ice, which is sparsely distributed in high-elevation mountainous areas of the United States, has unique and specific spectral characteristics, so it achieves 100% P.A. in the GLC_FCS30D. The large fluctuations in U.A. for ice and snow are attributed to: 1) the small sample size for ice and snow in the LCMAP_Val dataset, and 2) a few misclassified grass/bare land pixels are correctly identified as snow and ice during 2005-2014.

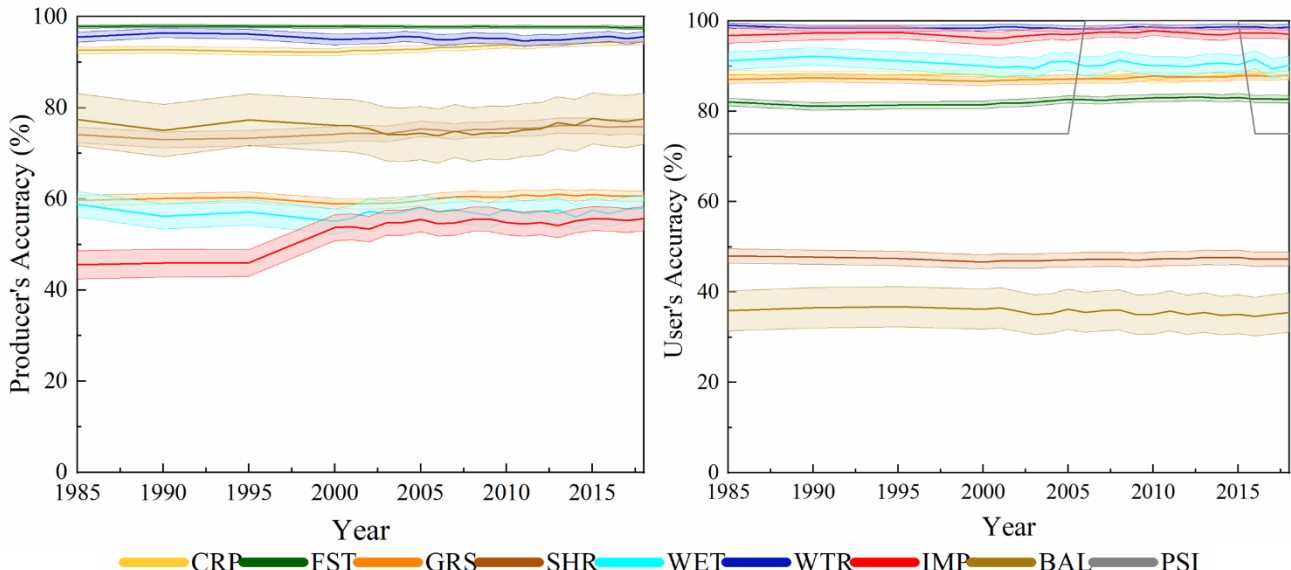

**Figure 10.** The time-series producer's accuracy and user's accuracy of GLC_FCS30D based on the LCMAP_Val dataset from 1985 to 2018 in the contiguous United States (CONUS). The error band represents ±1 standard errors.

Figure 11 indicates the area-bias percentage of eight land-cover types estimated by GLC_FCS30D and LCMAP_Val across the CONUS. Intuitively, the GLC_FCS30D and LCMAP_Val share similar total areas for estimation of cropland, bare land, and water body, and show evident area deviations for estimating forest, shrubland, and grassland. The deviations in shrubland and grassland are mainly because these land-cover types coexist in the semi-arid regions of the central United States and share similar spectral characteristics and temporal variability; thus, some grasslands in the LCMAP_Val are considered as shrubland in the GLC_FCS30D. Xian et al. (2022) also failed to distinguish grassland and shrubland and combined them as a group in generating the LCMAP annual maps. The LCMAP_Val has a broader definition of impervious surface and resulting negative bias, so the impervious surface area estimated in LCMAP_Val is larger than the assessment in GLC_FCS30D dataset.

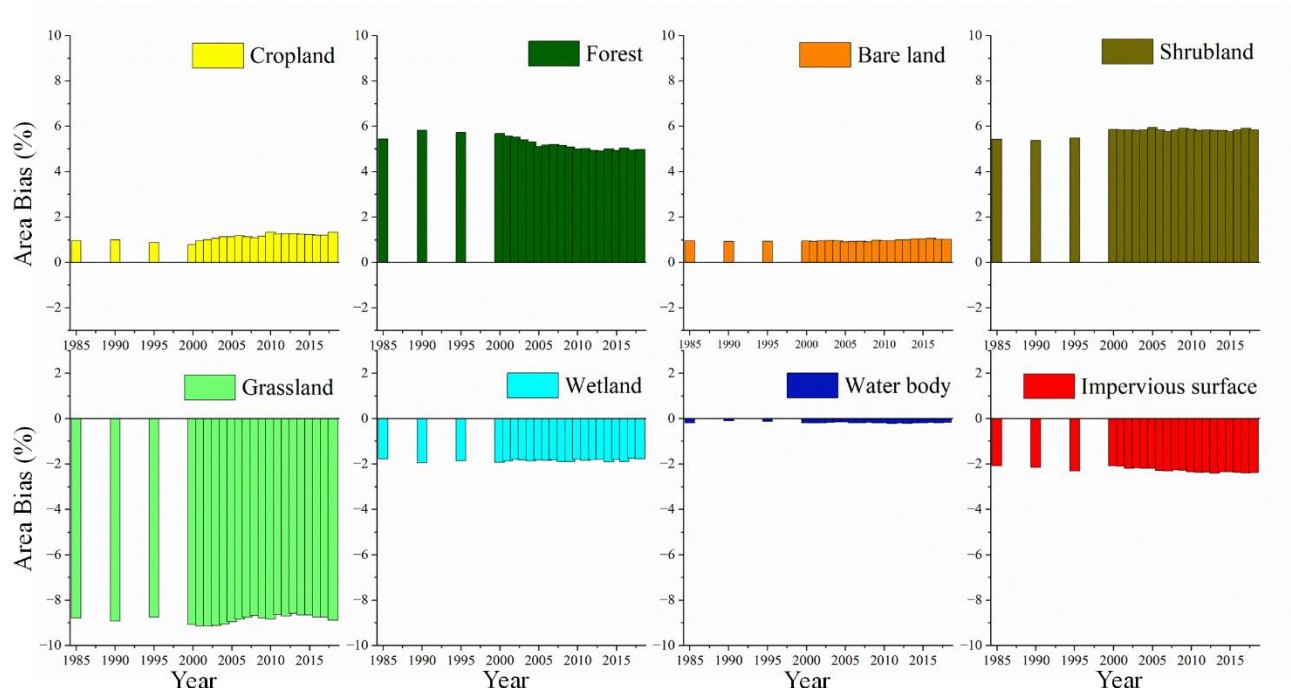

**Figure 11.** The area-bias percentage of eight land-cover types in the GLC_FCS30D and LCMAP_Val datasets from 1985 to 2017 in the contiguous United States (CONUS).

Table 4 further analyzed the confusion matrix of land-cover changed and unchanged pixels in
555 GLC_FCS30D using LCMAP_Val dataset. It should be noted that the land-cover changed samples in the LCMAP_Val was still sparse, that is, the size of changed samples cannot support the land-cover change analysis over specific land-cover changes. Similarly, Stehman et al. (2021) also grouped the land-cover types into 'No change' and 'change' types for analyzing the land-cover changes. In this study, using the 'changed' and 'unchanged' validation points in LCMAP_Val, the O.A. of the GLC_FCS30D reached the $90.49\pm0.45\%$. In
560 particular, the unchanged land-cover pixels played a dominant role and reached the high P.A. of 92.84% and U.A. of 96.28%. In contrast, the P.A. and U.A. of concerned land-cover changed pixels were $72.26\pm2.04\%$ and $56.62\pm2.00\%$, and its F1 score was 63.49%.

**Table 4**. The confusion matrix of changed and unchanged pixels in GLC_FCS30D using LCMAP_Val datasets.

|  | Unchanged | Changed | Total | P.A. (SE) | F1 |
|---|---|---|---|---|---|
| Unchanged | 82.21 | 6.34 | 88.55 | 92.84(0.42) | 94.53 |
| Changed | 3.18 | 8.27 | 11.45 | **72.26(2.04)** | 63.49 |
| Total | 85.39 | 14.61 |  |  |  |
| U.A. (SE) | 96.28(0.32) | **56.62(2.00)** |  |  |  |
| O.A. (SE) |  |  | 90.49(0.45) |  |  |

### 4.3.2 Time-series accuracy metrics of GLC_FCS30D from the LUCAS dataset

565 Table 5 lists the time-series accuracy metrics of the GLC_FCS30D dataset across the European Union (EU) from 2006 to 2018 using the LUCAS dataset. The GLC_FCS30D dataset has a mean O.A. of 81.91% (±0.09%) ranging from 81.64% (0.09%) to 82.11% (0.09%) in EU. The two dominant land-cover types (cropland and forest) that cover almost 70% of the entire EU area (Gao et al., 2020)) have higher P.A. and U.A. values than

other land-cover types. The P.A. and U.A. of cropland exceed 85% and 93%, respectively. Forest has unbalanced P.A. (approximately 95%) and U.A. values (approximately 76%) because the LUCAS dataset defines forest more broadly than the GLC_FCS30D dataset. In particular, sparse vegetation associated with forest is grouped as forest in LUCAS but as bare land in GLC_FCS30D. Gao et al. (2020) explained the discrepancy in the forest definition between LUCAS and GLC_FCS30. Shrubland, grassland, and bare land showed inferior performance in both P.A. and U.A. because of their complicated spectral variability and spatial heterogeneity. Gao et al. (2020) also found that three global 30-m land-cover products (GlobeLand30, FROM_GLC, and GLC_FCS30) exhibited poor performance for these three land-cover types. Urban green space and discontinuous urban fabric, excluding from the GLC_FCS30D, are grouped as impervious surface in the LUCAS. Thus, the impervious surface also has a low P.A. of approximately 59%. Last, we further investigate the temporal variability of P.A. and U.A. and find that permanent ice and snow and wetland show greater variability and that both are closely related to annual temperature and precipitation; namely, their spatial distributions are affected by the natural environment.

**Table 5.** Time-series accuracy metrics of the GLC_FCS30D dataset using the LUCAS validation dataset across the European Union.

| | 2006 | | 2009 | | 2012 | | 2015 | | 2018 | |
| --- | --- | --- | --- | --- | --- | --- | --- | --- | --- | --- |
| | P.A.(SE) | U.A.(SE) | P.A.(SE) | U.A.(SE) | P.A.(SE) | U.A.(SE) | P.A.(SE) | U.A.(SE) | P.A.(SE) | U.A.(SE) |
| CRP | 85.49(0.11) | 93.37(0.08) | 85.40(0.11) | 93.31(0.08) | 85.50(0.11) | 93.17(0.08) | 85.47(0.11) | 93.05(0.08) | 85.52(0.11) | 92.82(0.08) |
| FST | 95.22(0.08) | 76.71(0.15) | 94.97(0.08) | 76.71(0.15) | 94.79(0.09) | 76.82(0.15) | 94.36(0.09) | 76.82(0.15) | 93.71(0.09) | 76.85(0.15) |
| GRS | 6.13(0.26) | 21.31(0.83) | 6.10(0.26) | 21.13(0.83) | 6.05(0.26) | 20.98(0.83) | 6.08(0.26) | 20.71(0.82) | 5.99(0.26) | 20.74(0.82) |
| SHR | 8.13(0.42) | 8.93(0.46) | 8.25(0.43) | 8.92(0.46) | 8.02(0.42) | 8.77(0.46) | 7.84(0.42) | 8.60(0.45) | 8.35(0.43) | 8.96(0.46) |
| WET | 63.10(0.81) | 66.55 (0.81) | 61.40(0.81) | 65.55(0.82) | 61.86(0.81) | 66.21(0.82) | 62.64(0.81) | 66.60(0.81) | 62.94(0.81) | 65.34 (0.81) |
| WTR | 89.73(0.40) | 92.44(0.36) | 90.09(0.40) | 92.53(0.35) | 90.28(0.39) | 92.36(0.36) | 90.83(0.38) | 91.63(0.37) | 90.10(0.40) | 91.56(0.37) |
| IMP | 58.55(0.56) | 72.69(0.56) | 59.21(0.55) | 72.06(0.56) | 59.06(0.55) | 71.72(0.56) | 58.65(0.55) | 70.85(0.56) | 59.01(0.55) | 70.29(0.56) |
| BAL | 52.77(1.12) | 39.62(0.95) | 52.90(1.12) | 38.44(0.93) | 52.19(1.13) | 37.70(0.93) | 52.07(1.13) | 36.16(0.90) | 52.33(1.13) | 34.69(0.87) |
| PSI | 86.02(5.00) | 35.01(4.38) | 91.40(4.04) | 36.56(4.38) | 89.25(4.46) | 31.86(4.00) | 96.24(2.74) | 31.40(3.81) | 96.24(2.74) | 31.35(3.81) |
| O.A.(SE) | 82.11(0.09) | | 81.99(0.09) | | 81.97(0.09) | | 81.82(0.09) | | 81.64(0.09) | |

Table 6 shows the area proportions of 10 major land-cover types from the GLC_FCS30D dataset (Map) and LUCAS validation dataset (Ref), respectively. The area bias (AB) measures the area deviations of the two different datasets for the same land-cover type. Overall, the GLC_FCS30D overestimates the total area assessments of forest, bare land, and ice and snow, and underestimates the remaining land-cover types in comparison to the LUCAS estimations. In particular, the AB of forest is the most significant overestimation of +7.356%, and the underestimated MB of cropland is −4.086%. Cropland and forest cover together account for approximately 70% of the total EU area (Gao et al., 2020)), as a result, the area bias (AB) values for these two land-cover types are more noticeable or pronounced compared to the AB values of the other land-cover types.

**Table 6.** The area proportions and area bias (AB) values of 10 major land-cover types from the GLC_FCS30D dataset (Map) and the LUCAS validation dataset (Ref).

| | 2006 | | | 2009 | | | 2012 | | | 2015 | | | 2018 | | |
| --- | --- | --- | --- | --- | --- | --- | --- | --- | --- | --- | --- | --- | --- | --- | --- |
| | Map | Ref | AB | Map | Ref | AB | Map | Ref | AB | Map | Ref | AB | Map | Ref | AB |
| CRP | 46.48 | 50.62 | -4.14 | 46.46 | 50.64 | -4.18 | 46.59 | 50.67 | -4.08 | 46.63 | 50.69 | -4.06 | 46.77 | 50.74 | -3.97 |

| | | | | | | | | | | | | | | | |
|---|---|---|---|---|---|---|---|---|---|---|---|---|---|---|
| FST | 41.39 | 33.76 | 7.63 | 41.28 | 33.75 | 7.53 | 41.14 | 33.73 | 7.41 | 40.96 | 33.73 | 7.23 | 40.66 | 33.68 | 6.98 |
| GRS | 1.21 | 4.15 | -2.94 | 1.21 | 4.15 | -2.94 | 1.21 | 4.15 | -2.94 | 1.23 | 4.15 | -2.92 | 1.21 | 4.15 | -2.94 |
| SHR | 1.91 | 2.08 | -0.17 | 1.94 | 2.08 | -0.14 | 1.92 | 2.08 | -0.16 | 1.91 | 2.07 | -0.16 | 1.95 | 2.06 | -0.11 |
| WET | 1.70 | 1.75 | -0.05 | 1.68 | 1.74 | -0.06 | 1.68 | 1.73 | -0.05 | 1.69 | 1.71 | -0.02 | 1.73 | 1.72 | 0.01 |
| WTR | 2.75 | 2.85 | -0.1 | 2.76 | 2.85 | -0.09 | 2.77 | 2.85 | -0.08 | 2.81 | 2.85 | -0.04 | 2.79 | 2.86 | -0.07 |
| IMP | 3.18 | 3.82 | -0.64 | 3.25 | 3.83 | -0.58 | 3.25 | 3.82 | -0.57 | 3.27 | 3.82 | -0.55 | 3.32 | 3.82 | -0.5 |
| BAL | 1.32 | 0.95 | 0.37 | 1.36 | 0.95 | 0.41 | 1.37 | 0.95 | 0.42 | 1.42 | 0.95 | 0.47 | 1.49 | 0.95 | 0.54 |
| PSI | 0.06 | 0.02 | 0.04 | 0.06 | 0.02 | 0.04 | 0.07 | 0.02 | 0.05 | 0.07 | 0.02 | 0.05 | 0.07 | 0.02 | 0.05 |

Table 7 presented the confusion matrix of changed and unchanged pixels using the LUCAS validation datasets. The O.A. of the GLC_FCS30D reached $90.36 \pm 0.38\%$, the P.A. and U.A. of the changed pixels were $52.86 \pm 1.93\%$ and $73.31 \pm 1.74\%$, and the corresponding F1 score was 61.43%. In contrast, the unchanged land-cover pixels reached the high P.A. and U.A., and both two metrics exceeded 90%. Thus, the changed land-cover pixels were more difficult to capture comparing with these unchanged pixels. Similarly, Stehman et al. (2021) also found that the accuracy metrics of changed pixels were greatly lower than that of unchanged pixel, the producer's accuracy of changed pixels and unchanged pixels were 16% and 99%, respectively.

**Table 7.** The confusion matrix of changed and unchanged pixels in GLC_FCS30D using time-series LUCAS datasets across the Europe Union.

| | Unchanged | Changed | Total | P.A. (SE) | F1 |
|---|---|---|---|---|---|
| Unchanged | 82.69 | 2.79 | 85.48 | 96.73 | 94.49 |
| Changed | 6.84 | 7.68 | 14.52 | 52.86 | 61.43 |
| Total | 89.53 | 10.47 | | | |
| U.A. (SE) | 92.36(0.36) | 73.31(1.74) | | | |
| O.A. (SE) | | 90.36(0.38) | | | |

### 4.4 The comparisons with other global land-cover dynamic products

Figure 12 gave the qualitative comparisons between our GLC_FCS30D and two widely used land-cover dynamic datasets (CCI_LC and MCD12Q1) during 2001-2020 in the Indo-China Peninsula, in which experienced evident land-cover changes in forest deforestation and urban expansion. In terms of the urban expansion, three datasets revealed the quick urbanization in the mega-city of Bangkok, and the CCI_LC under-estimated the impervious surface areas in 2001 comparing with two other datasets. Meanwhile, the GLC_FCS30D also captured more spatial details (such as: rural building and road networks) than CCI_LC and MCD12Q1 because of its high spatial resolution of 30 m.

In terms of the most significant deforestation, the CCI_LC showed the worst performance because 1) it under-estimated the forest covers in the 2001 (the rectangle region 1, R1), that is, some forests were wrongly labeled as the croplands; 2) some deforested forests cannot be captured during the period of 2001-2020 in rectangle region 2 (R2), so their deforested forest area was less than that of GLC_FCS30D and MCD12Q1; and 3) there was obvious misclassification problem between forest and wetland in 2001 (the rectangle region 3, R3). Then, the MCD12Q1 also suffered the omission error for forest in R1, namely, the captured forest area in 2001 was lower than their actual areas based on the natural-color imagery. As for the evident deforestation in the R2, we can find that almost all forest pixels changed to the other land-cover types (savanna and grassland) in MCD12Q1, which was obviously deviated from the actual situation, thus, MCD12Q1 over-estimated the forest

deforestation. Meanwhile, the time-series MCD12Q1 showed various land-cover distributions in the R3, which indicated that the MCD12Q1 performed lower mapping accuracy and temporal stability for these wetland areas. In comparison, the GLC_FCS30D achieved the best performance in capturing the spatial distribution of forest in 2001, forest deforestation during 2001-2020, and wetland stability.

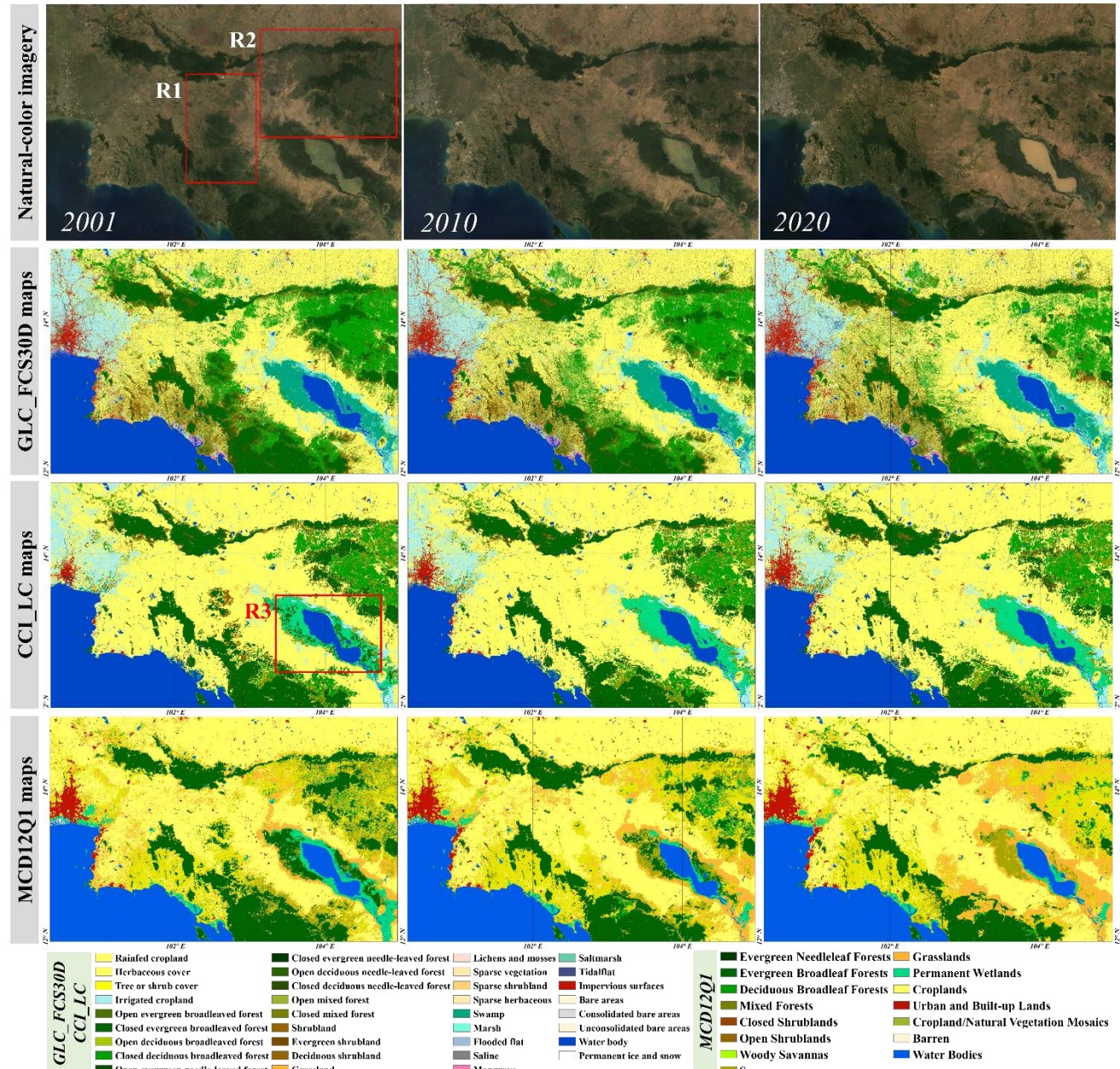

Figure 12. The comparisons between GLC_FCS30D with CCI_LC and MCD12Q1 land-cover dynamic products in Indo-China Peninsula during 2001-2020. The natural-color imagery are composited from the time-series Landsat imagery.

Figure 13 showed another comparison example about three datasets in Paraguay, South America, and the most evident land-cover change was the deforestation and cropland incensement according to the time-series natural-color Landsat imagery. In terms of the spatial distribution, the consistency between GLC_FCS30D and CCI_LC was higher, while the MCD12Q1 was obviously different from the other two datasets. A large amount of deciduous broadleaved forests were labeled as the savanna and woody savanna, and most croplands were

identified as the grasslands in the MCD12Q1, which mainly because of the difference of classification system. Then, as for the land-cover change areas, the GLC_FCS30D performed the highest accuracy and captured the richer spatial details. For example, the deforestation intensity during 2010-2020 was significantly greater than that during 2001-2010, and the GLC_FCS30D also revealed the regular deforestation caused by human factors. In contrast, the CCI_LC and MCD12Q1 failed to capture the deforestation during 2010-2020, and the small and fragmented changes (caused by human activities) also cannot be captured.

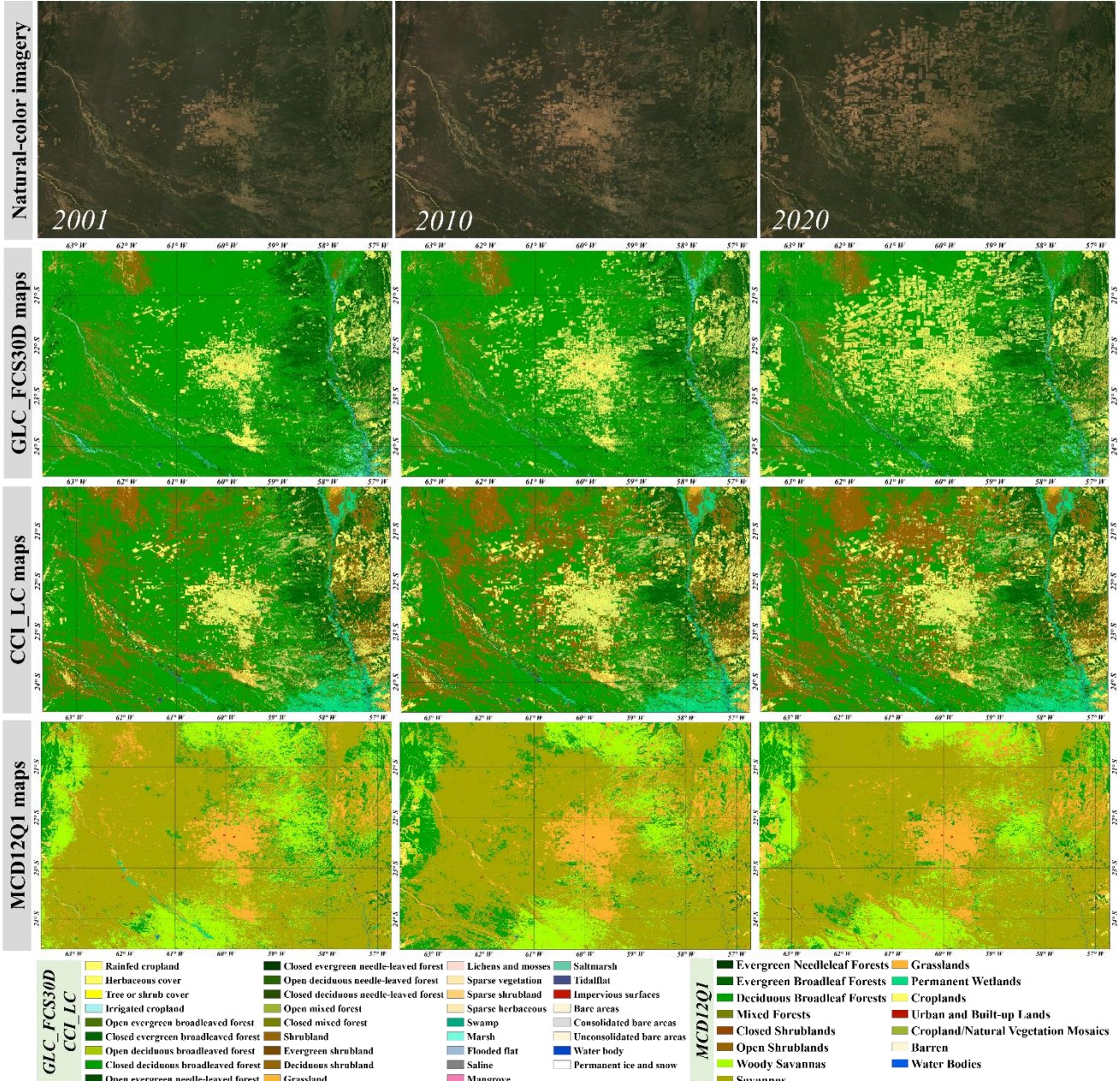

Figure 13. The comparisons between GLC_FCS30D with two time-series land-cover dynamic datasets in Paraguay, South America, during 2001-2020. The natural-color imagery are composited from the time-series Landsat imagery.

## 4.5 Limitations and perspectives of the GLC_FCS30D dataset

To achieve the goal of accurate and robust monitoring of global land-cover change, four steps are adopted: 1) combining the advantages of the CCD model and full time-series Landsat observations to capture the land-cover change time-points for any changed pixels; 2) using the temporally stable areas as prior knowledge to ensure the quality of training samples and adopting local adaptive modeling to update the land-cover transitions of these changed pixels; 3) independently developing global thematic products for two complicated land-cover types (impervious surface and wetland) to improve the reliability of the GLC_FCS30D; and 4) applying the 'spatiotemporal consistency checking' optimization in Section 3.3.3 to further guarantee the stability and accuracy of the GLC_FCS30D. The accuracy assessments, using the developed global validation dataset and two third-party datasets, demonstrate that the GLC_FCS30D fulfills accuracy requirements at a baseline year and for time-series variability over global or national scales. Comparisons with other land-cover products also highlight the superiority of the GLC_FCS30D in terms of classification system diversity and monitoring accuracy of these changed-areas. However, long time-series monitoring of global land-cover change is an extremely complex and difficult task (Hansen and Loveland, 2012; Song et al., 2018; Winkler et al., 2021; Xian et al., 2022). Although this study takes a series of measurements and methods to achieve global 30-m land cover change monitoring over past 37 years, there are still some uncertainties and limitations that need to be resolved in further work.

The CCD algorithm makes full use of dense satellite observations to capture land-cover changes robustly and accurately (Zhu and Woodcock, 2014b; Zhu et al., 2012). However, previous studies have demonstrated that their reliability is highly correlated to the density of valid satellite observations (Bullock et al., 2022; Ye et al., 2021; Zhu et al., 2019). Cloudy and snowy areas have greater uncertainty for capturing the time points of land-cover change (DeVries et al., 2015; Xian et al., 2022). Additionally, due to the limited storage capacity and satellite–ground data-transmission capacity of early satellites, the density of Landsat imagery is sparse before 2000 (only Landsat 5 single-satellite acquired data) (Roy et al., 2014b). In this study, we combine the satellite observations from two years before and after the nominal center year from 1985 to 1995; for example, we update the land-cover maps in 1995 using all available imagery from 1993 to 1997. However, a previous study found that northeastern Asia did not have any valid Landsat observations before 2000 (Zhang et al., 2022), which means some land-cover changes could not be captured in the GLC_FCS30D in these areas before 2000. To solve the problem of missing and sparse observations, a useful solution is to fuse multisourced remote-sensing imagery. For example, Zhang et al. (2021c) combined Landsat and Sentinel-2 imagery to track tropical forest disturbances with overall accuracy of more than 87%. Therefore, further work will investigate the feasibility of integrating Sentinel 1/2, SPOT, MODIS, and AVHRR imagery as auxiliary datasets to achieve the annual land-cover monitoring before 2000 and further ensure the land-cover monitoring quality.

To ensure stability of the GLC_FCS30D, a spatiotemporal consistency optimization algorithm, which was widely used in impervious surface change optimizations (Li et al., 2015; Zhang et al., 2022), was applied. It makes full use of the spatiotemporal neighbor pixels to calculate the land-cover homogeneity, and then remove the 'salt and pepper' noise caused by the pixel-based classifications. The qualitative comparisons in Amazon's deforestation areas and China's urban expansion areas (Figure S2 in the supplement material) also showed that the spatiotemporal consistency optimization can improve the data quality of GLC_FCS30D by suppressing 'salt and pepper' noise and optimizing the temporal consistency. Similarly, Yang and Huang (2021) used this algorithm to optimize China's annual land-cover products during 1999 during 1990-2019, and found that it improved the mapping accuracy of the time-series land-cover dataset.

GLC_FCS30D reveals a large number of land-cover changes in the semi-arid regions illustrated in Figure 5a, in which the land-cover changes are more influenced by climate factors. For example, the central region of Australia is a typical semi-arid region, and the dominant land-cover types are grassland, sparse vegetation, shrubland, and bare land. In general, if there is sufficient annual precipitation, the distributions of shrubland and grassland in the area will be more extensive; otherwise the area will be dominated by bare land and sparse vegetation (Dong et al., 2020; Ge et al., 2022). Recently, some studies suggested suppressing these changes; for example, Bastos et al. (2022) chose to suppress these land-cover changes by fusing these four land-cover types into the single grassland land-cover type in Australia, and Xian et al. (2022) combined grassland and shrubland together in the CONUS. Whether these frequent and climate-sensitive land-cover changes should be suppressed will be considered in our further work.

Although we used a global validation dataset to assess the capability of GLC_FCS30D in the baseline year of 2020 and two third-party regional datasets to assess its time-series accuracy variability in the European Union and the CONUS, the accuracy assessment work should be strengthened. In particular, the classification system differences among GLC_FCS30D, LUCAS, and LCMAP_Val cannot be ignored. For example, the impervious surface land-cover type in the LUCAS and LCMAP_Val contains artificial surfaces and their surroundings (such as city greenery) (Stehman et al., 2021; Xian et al., 2022), while the GLC_FCS30D only includes artificial structures (Zhang et al., 2022), so the impervious surface in GLC_FCS30D has low P.A. when validating with the LUCAS and LCMAP_Val datasets in Section 4.3. The time-series accuracy variability is only analyzed in two regions, so its performance in more complex areas (such as Africa and Asia) needs to be further investigated. Thus, our future work would be paid on long-term time-series validation data sets for more regions and on building a long time-series global validation dataset based on the existing works in Section 2.5.1, and then analyzed the accuracy metrics of the land-cover changed pixels for all land-cover type and their intra-annual variability.

## 5. Data availability

The developed GLC_FCS30D dataset can be freely accessible via https://doi.org/10.5281/zenodo.8239305 (Liu et al., 2023). To allow users to better select this dataset, it is saved as 961 5° × 5° independent tiles. Each tile is named as 'GLC_FCS30D_yyyyYYYY_E/W**N/S**.tif', in which 'E/W**N/S**' represents the longitude and latitude coordinates of the top-left corner, and yyyy and YYYY are the start and end years of the land-cover change monitoring. The GLC_FCS30D contains 26 time-step maps from 1985 to 2022, updated every five years before 2000 and annually from 2000 to 2022. It should be noted that the GLC_FCS30D adopted the 5-years cycle before 2000 because of the sparse availability of Landsat 5 imagery at early stage, thus, we sacrificed temporal cycle in guaranteeing the land-cover mapping accuracy. The first three time steps are saved together and the following 23 time steps are saved separately. For example, GLC_FCS30D_19851995_E115N15.tif and GLC_FCS30D_20002022_E115N15.tif are the first three time-steps and the following 23 annual time-steps data from 1985 to 2022 for the region of 115°–120°E, 10°–15°N, respectively.

## 6. Conclusion

Land cover change is the main cause or driving force of global climate change and has attracted increasing attention over the past decades. Long time-series global land-cover dynamic monitoring is still a challenging

task. In this study, the first global 30-m land-cover dynamic dataset with fine classification system (GLC_FCS30D), containing 35 fine land-cover sub-categories and covering the period of 1985 to 2022 with 26 time-steps, is generated on the GEE platform. In specific, we take advantage of the full time-series Landsat observations and the CCD algorithm to capture the time-points of changed areas, and then update and optimize the land-cover changed areas based on the local adaptive modeling strategy and a temporal-consistency algorithm. The accuracy assessments indicate that the proposed method can achieve accurate and spatiotemporally consistent land-cover change monitoring, and the GLC_FCS30D achieves an overall accuracy in 2020 of 80.88% ($\pm$0.27%) for the basic classification system 10 major land-cover types) and 73.04% ($\pm$0.30%) for the LCCS level-1 validation system (17 LCCS land-cover types). Therefore, the GLC_FCS30D is the first global land-cover dynamic monitoring product with a 37-year time span and the most diverse classification system. It will be essential for sustainable development, environmental protection, and informed decision-making to address the challenges of a rapidly changing world.

## Financial support

This research has been supported by the National Natural Science Foundation of China (grant no. 41825002), the Open Research Program of the International Research Center of Big Data for Sustainable Development Goals (grant no. CBAS2022ORP03) and the National Natural Science Foundation of China (grant no. 42201499).

## Author contributions

LL and XZ conceptualized and investigated the project. XZ, TZ, XC designed the methodology, TZ, WL, HX and JW performed the validation. XZ prepared the original draft of the paper, LL and HX reviewed and edited the paper.

## Competing interests

The authors declare that they have no conflict of interest.

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
