# Peer review of "GLC\_FCS30D: The first global 30-m land-cover dynamic monitoring product with a fine classification system from 1985 to 2022 using dense time-series Landsat imagery and continuous change-detection method"

_Earth System Science Data, 2023_

## Author Comment (AC1)

**Response to comments**

**Paper #:** essd-2023-320
**Title:** GLC_FCS30D: The first global 30-m land-cover dynamic monitoring product with a fine classification system from 1985 to 2022 using dense time-series Landsat imagery and continuous change-detection method
**Journal**: Earth System Science Data

**Reviewer #1**

This paper greatly attempts to map the annual global land cover change using all Landsat observations and the CCD approach. The paper is well written with clear logic, while some issues need to be clarified for publication as a scientific data paper.

Great thanks for the positive comments. The manuscript has been greatly improved based on your and another reviewers' comments and suggestions.

1. The CCDC approach is very sensitive to the parameters adopted, which may identify pseudo changes. This may directly relate to the derived temporal stable regions, and further impact the results of global changed areas (as well as the statistics). As a scientific data paper, this part should be enhanced with quantitative analysis, especially the uncertainty of change detection across different cover types.

Great thanks for the comment. Yes, we agree that the CCDC algorithm was sensitive to the parameter settings, thus the sensitive analysis between three sensitive parameters (minObservations, chiSquareProbability and minNumOfYearScaler) with the land-cover change detection accuracy (omission error and commission error) were supplemented, and the results showed that there was a correlation between the omission error and commission error with the variability of the parameter's values.

In specific, the descriptions of sensitive analysis and how to determine the values of the sensitive parameters have been added in the Section 3.2 as:

Next, the CCD was also a multi-parameter change detection model and demonstrated to be sensitive to the parameter settings (Xiao et al., 2023; Zhu and Woodcock, 2014b). The CCDC algorithm on the Google Earth Engine platform (ee.Algorithms.TemporalSegmentation.Ccdc) contained three key adjustable parameters: minObservations, chiSquareProbability and minNumOfYearScaler. Zhu et al. (2019) analyzed the relationships between the omission error and commission error of land-cover changes with the variability of three parameters in the United States, and found their values affected the change detection accuracy. In this study, we also investigated the sensitivity between parameter settings with the change detection accuracies in Figure S1 (seen the Supplement material) using the time-series points from LCMAP_Val and LUCAS datasets after partly sampling. Notably, the sensitivity analysis was implemented in two large-areas for ensure the feasibility of optimal parameters, that is, which will be suitable for other areas in land-cover change detection. The results also showed the CCD is a parameter-sensitive algorithm and the optimal parameter values were 5, 0.95 and 2-year for minObservations, chiSquareProbability and minNumOfYearScaler.

[Figure]

Figure S1. The sensitive analysis between the omission error and commission error with the minObservations, chiSquareProbability and minNumOfYearScaler using the time-series points from LCMAP_Val and LUCAS datasets after partly sampling.

Xiao, Y., Wang, Q., Tong, X., and Atkinson, P. M.: Thirty-meter map of young forest age in China, Earth Syst. Sci. Data, 15, 3365-3386, https://doi.org/10.5194/essd-15-3365-2023, 2023.

Zhu, Z., Zhang, J., Yang, Z., Aljaddani, A. H., Cohen, W. B., Qiu, S., and Zhou, C.: Continuous monitoring of land disturbance based on Landsat time series, Remote Sensing of Environment, https://doi.org/10.1016/j.rse.2019.03.009, 2019.

Meanwhile, using the values were 5, 0.95 and 2 years for minObservations, chiSquareProbability and minNumOfYearScaler, we also analyzed the uncertainty of the CCD algorithm at various land-cover change types based on the random sampling land-cover changed points from NLCD products over the United States in Table S1.

Table S1. The uncertainty analysis of CCD algorithm for different land-cover change types using the random sampling land-cover changed points from NLCD products over the United States.

|  | To cropland | To forest | To shrubland | To grassland | To water | To bareland | To impervious surface |
|---|---|---|---|---|---|---|---|
| P.A. (±SE) | 70.00(0.55) | 45.64(0.49) | 42.02(0.89) | 66.13(0.33) | 77.05(0.54) | 42.86(1.32) | 70.65(0.42) |
| U.A. (±SE) | 46.10(0.48) | 64.66(0.56) | 23.81(0.56) | 43.94(0.28) | 58.02(0.55) | 35.29(1.16) | 57.33(0.42) |

It should be noted that the wetland was independently generated, and the permanent snow ice and tundra were sparse distributed on the United States, the uncertainty analysis of CCD for these three classes were excluded.

2. As a global land cover change dataset, the accuracy assessment regarding those changed pixels should be significantly improved. The confusion matrix just shows the accuracy in 2020, while those changes in different years with field samples

Great thanks for the comment. Yes, we completely agree that the accuracy analyzing for those changed pixels should be taken care. However, collecting a time-series global validation datasets are greatly difficult and challenging, and **there is currently no public long time-series validation datasets**.

In this revision, the time-series LCMAP_Val and LUCAS validation points are further used to calculate the accuracy metrics of changed and unchanged pixels. It should be noted that 'changed' and 'unchanged' referred

to the work of Stehman et al. (2021) in validating the LCMAP annual land cover products during 1985-2017. The specific analysis about land-cover changed pixels are added as:

**Section 3.4 Accuracy assessment**
Lastly, to quality the performance of land-cover changed pixels, we obtained the proposal of Stehman et al. (2021) in assessing the LCMAP annual land-cover products 1985-2017, that is, the validation pixels were grouped into "changed" and "unchanged" categories and the corresponding confusion matrix were calculated. Meanwhile, to minimize the imbalance in the sample size of "change" and "no-change" samples, the metrics of F1 score was supplemented as:

$$F1 = \frac{P.A. \times U.A.}{P.A. + U.A.} \times 2 \times 100\% \tag{5}$$

**Section 4.3 Accuracy assessment based on two third-party regional validation datasets**

Table 4 further analyzed the confusion matrix of land-cover changed and unchanged pixels in GLC_FCS30D using LCMAP_Val dataset. It should be noted that the land-cover changed samples in the LCMAP_Val was still sparse, that is, the size of changed samples cannot support the land-cover change analysis over specific land-cover changes. Similarly, Stehman et al. (2021) also grouped the land-cover types into 'No change' and 'change' types for analyzing the land-cover changes. In this study, using the 'changed' and 'unchanged' validation points in LCMAP_Val, the O.A. of the GLC_FCS30D reached the 90.49±0.45%. In particular, the unchanged land-cover pixels played a dominant role and reached the high P.A. of 92.84% and U.A. of 96.28%. In contrast, the P.A. and U.A. of concerned land-cover changed pixels were 72.26±2.04% and 56.62±2.00%, and its F1 score was 63.49%.

**Table 4**. The confusion matrix of changed and unchanged pixels in GLC_FCS30D dataset using LCMAP_Val datasets.

|  | Unchanged | Changed | Total | P.A. (SE) | F1 |
|---|---|---|---|---|---|
| Unchanged | 82.21 | 6.34 | 88.55 | 92.84(0.42) | 94.53 |
| Changed | 3.18 | 8.27 | 11.45 | **72.26(2.04)** | 63.49 |
| Total | 85.39 | 14.61 |  |  |  |
| U.A. (SE) | 96.28(0.32) | **56.62(2.00)** |  |  |  |
| O.A. (SE) |  | 90.49(0.45) |  |  |  |

Table 7 presented the confusion matrix of changed and unchanged pixels using the LUCAS validation datasets. The overall accuracy of the GLC_FCS30D reached 90.36±0.38%, the P.A. and U.A. of the changed pixels were 52.86±2.04% and 73.31±2.00%, and the corresponding F1 score was 61.43%. In contrast, the unchanged land-cover pixels reached the high P.A. and U.A., and both two metrics exceeded 90%. Thus, the changed land-cover pixels were more difficult to capture comparing with these unchanged areas. Similarly, Stehman et al. (2021) also found that the accuracy metrics of changed pixels were greatly lower than that of unchanged pixel.

**Table 7.** The confusion matrix of changed and unchanged pixels in GLC_FCS30D using time-series LUCAS datasets across the Europe Union.

|  | Unchanged | Changed | Total | P.A. (SE) | F1 |
|---|---|---|---|---|---|
| Unchanged | 82.69 | 2.79 | 85.48 | 96.73 | 94.49 |

| | Changed | 6.84 | 7.68 | 14.52 | 52.86 | 61.43 |
|---|---|---|---|---|---|---|
| | Total | 89.53 | 10.47 | | | |
| | U.A. (SE) | 92.36(0.36) | 73.31(1.74) | | | |
| | O.A. (SE) | | 90.36(0.38) | | | |

Stehman, S. V., Pengra, B. W., Horton, J. A., and Wellington, D. F.: Validation of the U.S. Geological Survey's Land Change Monitoring, Assessment and Projection (LCMAP) Collection 1.0 annual land cover products 1985–2017, Remote Sensing of Environment, 265, 112646, https://doi.org/10.1016/j.rse.2021.112646, 2021.

Meanwhile, we also highlighted the necessity of strengthening the accuracy analysis for land-cover changed pixels in our future works in Section 4.5 as:

The time-series accuracy variability is only analyzed in two regions, so its performance in more complex areas (such as Africa and Asia) needs to be further investigated. Thus, future work would be paid on collecting long-term time-series validation data sets for more regions, and on building a long time-series global validation dataset, and then analyzed the accuracy metrics of the land-cover changed pixels for all land-cover types.

As well as global product comparison (e.g., ESACCI and MODIS) should be given and discussed.

Based on your suggestion, the comparisons with ESACCI and MCD12Q1 have been added in two typical areas in Section 4.4 as:

**4.4 The comparisons with other global land-cover dynamic products**

Figure 12 gave the qualitative comparisons between our GLC_FCS30D and two widely used land-cover dynamic datasets (CCI_LC and MCD12Q1) during 2001-2020 in the Indo-China Peninsula, in which experienced evident land-cover changes in forest deforestation and urban expansion. In terms of the urban expansion, three datasets revealed the quick urbanization in the mega-city of Bangkok, and the CCI_LC under-estimated the impervious surface areas in 2001 comparing with two other datasets. Meanwhile, the GLC_FCS30D also captured more spatial details (such as: rural building and road networks) than CCI_LC and MCD12Q1 because of its high spatial resolution of 30 m.

As the most significant deforestation, the CCI_LC showed the worst performance because 1) it under-estimated the forest covers in the 2001 (the rectangle region 1), that is, some forests were wrongly labeled as the croplands; 2) some deforested forests cannot be captured during the period of 2001-2020 in rectangle region 2 (R2), and their deforested forest area was less than that of GLC_FCS30D and MCD12Q1; and 3) there was obvious misclassification problem between forest and wetland in 2000 (the rectangle region 3, R3). Then, the MCD12Q1 also suffered the omission error for forest in rectangle region 1, namely, the captured forest area in 2000 was lower than their actual areas in natural-color imagery. As for the evident deforestation in the region 2, we can find that almost all forest pixels changed to the other land-cover types (savanna and grassland), which was obviously deviated from the actual situation, thus, MCD12Q1 over-estimated the forest deforestation. Meanwhile, the time-series MCD12Q1 showed various land-cover distributions in the R3, which indicated that the MCD12Q1 performed lower mapping accuracy and temporal stability for these wetland areas. In comparison, the GLC_FCS30D achieved the best performance in capturing the spatial distribution of forest in 2000, forest deforestation during 2001-2020, and wetland stability.

[Figure]

Figure 12. The comparisons between GLC_FCS30D with CCI_LC and MCD12Q1 land-cover dynamic products in Indo-China Peninsula during 2001-2020. The natural-color imagery are composited from the time-series Landsat imagery.

Figure 13 showed another comparison example about three datasets in Paraguay, South America, and the most evident land-cover change was the deforestation and cropland incensement according to the time-series natural-color Landsat imagery. In terms of the spatial distribution, the consistency between GLC_FCS30D and CCI_LC was higher, while the MCD12Q1 was obviously different from the other two datasets. A large amount of deciduous broadleaved forests were labeled as the savanna and woody savanna, and most croplands were identified as the grasslands in the MCD12Q1, which mainly because of the difference of classification system. Then, as for the land-cover change areas, the GLC_FCS30D performed the highest accuracy and captured the richer spatial details. For example, the deforestation intensity during 2010-2020 was significantly greater than that during 2001-2010, and the GLC_FCS30D also revealed the regular deforestation caused by human factors.

In contrast, the CCI_LC and MCD12Q1 failed to capture the deforestation during 2010 and 2020, and the small and fragmented changes (caused by human activities) also cannot be captured.

[Figure]

Figure 13. The comparisons between GLC_FCS30D with two time-series land-cover dynamic datasets in Paraguay, South America, during 2001-2020.

---

## Author Comment (AC2)

**Response to comments**

**Paper #:** essd-2023-320
**Title:** GLC_FCS30D: The first global 30-m land-cover dynamic monitoring product with a fine classification system from 1985 to 2022 using dense time-series Landsat imagery and continuous change-detection method
**Journal**: Earth System Science Data

**Reviewer #2**

The authors present a very detailed manuscript on the generation of a global 30-m land cover product. It is original in the spatio-temporal density of Landsat satellite imagery used to generate annual maps over nearly 30 years. I recommend this paper for publication. There is one major issue that needs to be addressed and a number of minor or editorial issues to address:

Great thanks for the positive comments. The manuscript has been further improved based on your and another reviewers' comments and suggestions.

**Major change:**
Both the Continuous Change Detection and Classification (CCDC) Algorithm and the Random Forest Algorithm for subsequent land cover classification use several hyperparameters. Both models will be sensitive to the hyperparameters selected. As a minimum, the hyperparameters selected for both models need to be clearly defined and justified (this is already partially done for the CCDC algorithm). However, to fully justify the use of hyperparameters, sensitivity analysis should be provided of the values used, and validation that the optimum or a favorable set of hyperparameters values have been selected.

Great thanks for the comment. Based on your suggestion and another reviewer's comment, the analysis of how to determine the parameters of CCDC algorithm have been added in the Section 3.2 as:

Next, the CCD was also a multi-parameter change detection model and demonstrated to be sensitive to the parameter settings (Xiao et al., 2023; Zhu and Woodcock, 2014b). The CCDC algorithm on the Google Earth Engine platform (ee.Algorithms.TemporalSegmentation.Ccdc) contained three key adjustable parameters: minObservations, chiSquareProbability and minNumOfYearScaler. Zhu et al. (2019) analyzed the relationships between the omission error and commission error of land-cover changes with the variability of three parameters in the United States, and found their values affected the change detection accuracy. In this study, we also investigated the sensitivity between parameter settings with the change detection accuracies in Figure S1 (seen the Supplement material) using the time-series points from LCMAP_Val and LUCAS datasets after partly sampling. Notably, the sensitivity analysis was implemented in two large-areas for ensure the feasibility of optimal parameters, that is, which will be suitable for other areas in land-cover change detection. The results also showed the CCD is a parameter-sensitive algorithm and the optimal parameter values were 5, 0.95 and 2-year for minObservations, chiSquareProbability and minNumOfYearScaler.

[Figure]

Figure S1. The sensitive analysis between the omission error and commission error with the minObservations, chiSquareProbability and minNumOfYearScaler using the time-series points from LCMAP_Val and LUCAS datasets after partly sampling.

Xiao, Y., Wang, Q., Tong, X., and Atkinson, P. M.: Thirty-meter map of young forest age in China, Earth Syst. Sci. Data, 15, 3365-3386, https://doi.org/10.5194/essd-15-3365-2023, 2023.

Zhu, Z., Zhang, J., Yang, Z., Aljaddani, A. H., Cohen, W. B., Qiu, S., and Zhou, C.: Continuous monitoring of land disturbance based on Landsat time series, Remote Sensing of Environment, https://doi.org/10.1016/j.rse.2019.03.009, 2019.

As for the parameters of random forest classifier, it only contains two adjustable parameters (the number of decision tree (Ntree) and predicted variables (Mtry)), and many previous studies have quantitatively or theoretically analyzed the influence of the parameters on the classification accuracy, and found that the classification accuracy was less sensitive to the selection of Ntree and Mtry. Thus, the default recommended setting of 500 and the square of the total number of input features were used. Correspondingly, the descriptions of how to determine these two parameters have been added in the manuscript as:

Thus, the RF algorithm was used to combine the training samples and multisourced features for updating the changed pixels. The RF algorithm allows for adjusting two key parameters **(the number of decision tree (Ntree) and predicted variables (Mtry)), and previous studies have quantitatively investigated the relationships between classification accuracy with the settings of these two parameters. Both theoretical and experimental results indicated that the selection of Mtry and Ntree had little influence on the classification accuracy (Belgiu and Drăguţ, 2016; Du et al., 2015). Thus, the default recommended values of 500 for Ntree and the square of the total number of input features for Mtry were used based on previous studies (Belgiu and Drăguţ, 2016; Zhang et al., 2019).**

**Minor or editorial changes:**

1. Introduction/ methods – various mentions of model 'accuracy' is used. This includes, but is not limited to lines 31-32, 69 and 155. Please be specific on the accuracy metric(s) used.

Great thanks for the comment. The 'overall accuracy' in the whole manuscript (line 31-32, 69 and 155) is an accuracy metric in the confusion matrix. In this manuscript, three accuracy metrics including: overall accuracy (O.A.), producer's accuracy (P.A.) and user's accuracy (U.A.) have been used and the corresponding formulas are also added in the Section 3.4 (accuracy assessment) as:

The validation process for the GLC_FCS30D dataset follows the recommended guidelines proposed by Pontus Olofsson (2014). These guidelines encompass two key components: area estimation (nonsite-specific accuracy) and accuracy assessment (site-specific accuracy). The site-specific accuracy assessment mainly focuses on estimating the confusion matrix and calculating some accuracy metrics including overall accuracy (O.A.),

producer's accuracy (P.A.), user's accuracy (U.A.) and the corresponding standard errors using a poststratified estimator (Pontus Olofsson, 2014).

$$P.A._{\cdot k} = \frac{p_{kk}}{\sum p_{k\cdot}}, U.A._{\cdot k} = \frac{p_{kk}}{\sum p_{\cdot k}}, O.A. = \sum_{k=1}^{m} p_{kk} \tag{4}$$

Where $p_{kk}$ was the proportion of the area mapped as class $k$ that had reference class $k$, $\sum p_{k\cdot}$ and $\sum p_{\cdot k}$ were the proportion of the area mapped as class $k$ and the proportion of the reference area as class $k$, and the $m$ denoted the number of land-cover types. Afterwards, because there is currently no global long-time series validation dataset, we used 84526 global validation points to assess the accuracy metrics of the GLC_FCS30D dataset in 2020 and used two third-party datasets to analyze the time-series accuracy variations. The GLC_FCS30D adopts a fine classification system containing 35 subcategories, for which we applied an analysis protocol into two validation systems (the level-0 classification system containing 10 major land-cover types and the LCCS level-1 validation system containing 17 land-cover types) to comprehensively understand the GLC_FCS30D dataset quality. The relationship between Level-0 and LCCS level-1 validation systems is explained in Table 1. Lastly, to quality the performance of land-cover changed pixels, we followed the proposal of Stehman et al. (2021) in assessing the LCMAP annual land-cover products 1985-2017, that is, the validation pixels were grouped into "changed" and "unchanged" categories and the corresponding confusion matrix were calculated. Meanwhile, to minimize the imbalance in the sample size of "change" and "no-change" samples, the metrics of F1 score was supplemented as:

$$F1 = \frac{P.A. \times U.A.}{P.A. + U.A.} \times 2 \times 100\% \tag{5}$$

2. Line 24- The use of the phrase 'In specific' is awkward and I suggest changing e.g. 'Specifically' or 'In particular…'
Great thanks for the suggestion. The 'In specific' has been changed as 'Specifically' and 'In particular' through the whole manuscript.

3. Line 201- 'The first time series validation set was assessed the performance…' remove 'was'.
Great thanks for pointing out the mistake. It has been corrected.

4. Line 205- Change 'It developed by combining…' to 'It was developed by combining…'
Great thanks for pointing out the mistake. It has been corrected.

5. Figure 2: This Figure is very useful for help the reader understand the main processes carried out in this project. Please add the shortened names of each dataset to the flow chart to make it even easier for the reader to follow the text.
Great thanks for the comment. Based on your suggestion, the shortened names of each dataset have been added and also bolded into the flowchart as:

[Figure]

6. Figure 2: You refer to masking 'poor quality' pixels. Please be more specific on this.

Great thanks for the comment. The 'poor quality' refers to these cloud, shadow and saturated pixels, as well as the Scan Line Corrector Off pixels in Landsat 7, which was added in the revision version.

Does this just include applying a cloud mask, or does it also consider issues with the Scan Line Corrector on Landsat 7, for example. What cloud mask was used.

Yes, the Scan Line Corrector Off pixels are also masked. Specifically, the 'poor-quality' pixels were masked using the CFmask algorithm, which was demonstrated to achieve high accuracy and great robustness for masking these 'poor-quality' pixels.

How did you account for pixels that may be under light cloud/ haze which may not be picked up by a cloud mask (e.g. does the CCDC intend to overcome this?)

In terms of these light cloud/haze pixels, actually, the Tmask algorithm, which was integrated into the CCDC algorithm in the GEE platform, was used to further minimize their effects.

**The explanations have been added in Section 3 as:**

Before detecting the land-cover changed pixels, all 'poor quality' pixels (cloud, shadow and saturated pixels, as well as the Scan Line Corrector Off pixels in Landsat 7) in the continuous time-series Landsat imagery were firstly masked using the CFmask algorithm, which was demonstrated to achieve the overall accuracy of 96.4% and was adopted by the USGS as official cloud- and shadow detection algorithm (Zhu et al., 2015; Zhu and Woodcock, 2012). Then, in terms of these residual cloud pixels (light cloud and haze contaminated pixels), the Tmask (multiTemporal mask) algorithm, which used the temporal information from these clear-sky pixels to improve the cloud-detection capability (Zhu and Woodcock, 2014a), was used to mask the residual cloud pixels. It should be noted that the Tmask has been integrated into the CCD algorithm on the GEE platform as ee.Algorithms.TemporalSegmentation.Ccdc(), that is, the effect of 'poor-quality' pixels were minimized.

7. Table 1: Please add the abbreviations for each land cover type to this table (at later points you refer to Table 1 as containing these).

Great thanks for the suggestion. The abbreviations of each land-cover type have been into the Table 1 as:

| Basic classification system | | Level-1 validation system | | Fine classification system | Id |
|---|---|---|---|---|---|
| Cropland | CRP | Rainfed cropland | RCP | Rainfed cropland | 10 |

| | | | | Herbaceous cover cropland | 11 |
|---|---|---|---|---|---|
| | | | | Tree or shrub cover cropland | 12 |
| | | Irrigated cropland | ICP | Irrigated cropland | 20 |
| Forest | FST | Evergreen broadleaved forest | EBF | Closed evergreen broadleaved forest | 51 |
| | | | | Open evergreen broadleaved forest | 52 |
| | | Deciduous broadleaved forest | BDF | Closed deciduous broadleaved forest | 61 |
| | | | | Open deciduous broadleaved forest | 62 |
| | | Evergreen needleleaved forest | ENF | Closed evergreen needleleaved forest | 71 |
| | | | | Open evergreen needleleaved forest | 72 |
| | | Deciduous needleleaved forest | DNF | Closed deciduous needleleaved forest | 81 |
| | | | | Open deciduous needleleaved forest | 82 |
| | | Mixed-leaf forest | MFT | Closed mixed-leaf forest | 91 |
| | | | | Open mixed-leaf forest | 92 |
| Shrubland | SHR | Shrubland | SHR | Shrubland | 120 |
| | | | | Evergreen shrubland | 121 |
| | | | | Deciduous shrubland | 122 |
| Grassland | GRS | Grassland | GRS | Grassland | 130 |
| Tundra | TUD | Lichens and mosses | LMS | Lichens and mosses | 140 |
| Wetland | WET | Inland wetland | IWL | Swamp | 181 |
| | | | | Marsh | 182 |
| | | | | Flooded flat | 183 |
| | | | | Saline | 184 |
| | | Coastal wetland | CWL | Mangrove | 185 |
| | | | | Salt marsh | 186 |
| | | | | Tidal flat | 187 |
| Impervious surface | IMP | Impervious surface | IMP | Impervious surface | 190 |
| Bare areas | BAL | Sparse vegetation | SVG | Sparse vegetation | 150 |
| | | | | Sparse shrubland | 152 |
| | | | | Sparse herbaceous cover | 153 |
| | | Bare areas | BAL | Bare areas | 200 |
| | | | | Consolidated bare areas | 201 |
| | | | | Unconsolidated bare areas | 202 |
| Water body | WTR | Water body | WTR | Water body | 210 |
| Permanent snow and ice | PSI | Permanent snow and ice | PSI | Permanent snow and ice | 220 |

8. Line 357- please provide more information on the indicator function, or at least a reference.

Great thanks for the comment. The description of the indicator function has been strength and the reference is also added as:

"and the I() denotes the indicator function for the equation of the status between two pixels. Namely, if $L_{x',y',t'}$ was equal to the $L_{x,y,t}$, then the value of indicator function was 1, otherwise it was equal to 0 (Kenny, 2003)"

Kenny, Q. Y.: Indicator function and its application in two-level factorial designs, The Annals of Statistics, 31, 984-994, https://doi.org/10.1214/aos/1056562470, 2003.

9. Figure 7- the very thick lines corresponding to pixels with a stable land cover overwhelm this image and make it difficult for the reader to decipher the most dominant types of land cover change. Please either rescale the image or consider removing the lines corresponding to no land cover change to make it easier for the reader to assess the dominant types of land cover change.

Great thanks for the comment. To make the Sankey diagram clear, the layout has been changed. The updated Figure 7 are following:

[Figure]

**Figure 7**. Sankey diagrams of the global land-cover changes during 1985-2022 in the GLC_FCS30D dataset.

10. Table 2 and 3- please use a method to highlight the relative performance of your algorithms. For example use a colour ramp or make particular values bold.

Great thanks for the comment. Since the ESSD journal does not allow colormaps to be added to the Table, some particular accuracy values (mentioned in the manuscript) have been bolded in Table 2 and 3 based on your suggestion as:

**Table 2.** Error matrix of the GLC_FCS30D dataset in 2020 based on the level-0 basic classification system. The reported Producer's Accuracy (P.A.) and User's Accuracy (U.A.) come with their corresponding standard errors (SE) shown in parentheses.

| | Map | | | | | | | | | | | |
|---|---|---|---|---|---|---|---|---|---|---|---|---|
| | | | | | **O.A. = 80.88% (±0.27%)** | | | | | | | |
| Reference | CRP | FST | GRS | SHR | WET | WTR | TUD | IMP | BAL | PSI | Total | P.A.(SE) |
| CRP | 15.442 | 0.792 | 0.679 | 0.388 | 0.086 | 0.027 | 0 | 0.174 | 0.117 | 0 | 17.704 | 87.22(0.54) |
| FST | 0.513 | 28.712 | 0.315 | 0.811 | 0.371 | 0.021 | 0.008 | 0.063 | 0.113 | 0.002 | 30.93 | **92.83(0.31)** |
| GRS | 1.035 | 1.166 | 5.906 | 1.181 | 0.231 | 0.011 | 0.084 | 0.051 | 1.181 | 0.01 | 10.855 | **54.41(1.02)** |
| SHR | 0.555 | 1.798 | 0.863 | 5.392 | 0.161 | 0.013 | 0.019 | 0.05 | 0.502 | 0.002 | 9.356 | **57.63(1.09)** |
| WET | 0.068 | 0.465 | 0.156 | 0.157 | 4.047 | 0.347 | 0.031 | 0.021 | 0.222 | 0.001 | 5.516 | 73.37(1.27) |
| WTR | 0.04 | 0.086 | 0.019 | 0.017 | 0.302 | 3.305 | 0.008 | 0.012 | 0.039 | 0.002 | 3.831 | 86.28(1.12) |
| TUD | 0.01 | 0.123 | 0.168 | 0.167 | 0.018 | 0.03 | 2.444 | 0.002 | 0.473 | 0.02 | 3.454 | 70.76(1.65) |
| IMP | 0.084 | 0.058 | 0.024 | 0.04 | 0.001 | 0.006 | 0.002 | 5.043 | 0.024 | 0 | 5.283 | 95.45(0.61) |
| BAL | 0.13 | 0.049 | 0.783 | 0.585 | 0.043 | 0.045 | 0.577 | 0.048 | 9.239 | 0.131 | 11.628 | 79.45(0.8) |
| PSI | 0 | 0.004 | 0.03 | 0.005 | 0 | 0.023 | 0.001 | 0 | 0.03 | 1.351 | 1.443 | 93.63(1.38) |
| Total | 17.877 | 33.251 | 8.943 | 8.743 | 5.259 | 3.828 | 3.176 | 5.464 | 11.94 | 1.52 | | |
| U.A.(SE) | 86.38 (0.55) | 86.35 (0.4) | 66.05 (1.07) | 61.68 (1.11) | 76.96 (1.2) | 86.33 (1.35) | 76.97 (1.6) | 92.29 (0.77) | 77.38 (0.82) | 88.89 (1.72) | | |

**Table 3.** Error matrix of the GLC_FCS30D dataset in 2020 based on the LCCS level-1 validation system. The reported Producer's Accuracy (P.A.) and User's Accuracy (U.A.) come with their corresponding standard errors (SE) shown in parentheses.

| Reference | RCP | ICP | EBF | DBF | ENF | DNF | MFT | SHR | GRS | LMS | SVG | IWL | CWL | IMP | BAL | WTR | PSI | Total | P.A. (SE) |
|---|---|---|---|---|---|---|---|---|---|---|---|---|---|---|---|---|---|---|---|
| RCP | 12.225 | **1.023** | 0.239 | 0.358 | 0.102 | 0.016 | 0.009 | 0.382 | 0.66 | 0 | 0.078 | 0.056 | 0.005 | 0.124 | 0.028 | 0.001 | 0 | 15.332 | 79.7(0.7) |
| ICP | 0.397 | 1.932 | 0.026 | 0.016 | 0.005 | 0 | 0 | 0.01 | 0.025 | 0 | 0.012 | 0.029 | 0.005 | 0.052 | 0 | 0.018 | 0 | 2.527 | 76.45(1.81) |
| EBF | 0.2 | 0.048 | 9.091 | 1.098 | 0.262 | 0.103 | **0.151** | 0.371 | 0.084 | 0 | 0.012 | 0.136 | 0.028 | 0.029 | 0.001 | 0.004 | 0 | 11.514 | 78.96(0.82) |
| DBF | 0.187 | 0.016 | 0.632 | 6.838 | 0.537 | 0.294 | **0.396** | 0.235 | 0.144 | 0.002 | 0.019 | 0.077 | 0.002 | 0.025 | 0.005 | 0.004 | 0.002 | 9.054 | 75.53(0.97) |
| ENF | 0.046 | 0.004 | 0.174 | 0.316 | 5.681 | 0.328 | **0.439** | 0.128 | 0.034 | 0.006 | 0.043 | 0.094 | 0 | 0.008 | 0.01 | 0.01 | 0 | 6.895 | **82.39(0.98)** |
| DNF | 0.008 | 0 | 0.002 | 0.13 | 0.245 | 1.854 | **0.073** | 0.071 | 0.053 | 0 | 0.011 | 0.025 | 0 | 0.001 | 0.007 | 0.002 | 0 | 2.414 | 76.79(1.85) |
| MFT | 0.004 | 0 | **0.019** | 0.176 | 0.234 | 0.013 | 0.828 | 0.014 | 0.004 | 0 | 0 | 0.010 | 0.05 | 0 | 0.001 | 0 | 0 | 1.308 | **58.29(1.53)** |
| SHR | 0.518 | 0.042 | 0.299 | 0.9 | 0.328 | 0.131 | 0.034 | 5.44 | 0.871 | 0.019 | 0.441 | 0.157 | 0.005 | 0.05 | 0.065 | 0.013 | 0.002 | 9.438 | 57.63(1.09) |
| GRS | 0.947 | 0.097 | 0.167 | 0.582 | 0.209 | 0.154 | 0.024 | 1.191 | 5.958 | 0.085 | 0.974 | 0.229 | 0.006 | 0.052 | 0.217 | 0.008 | 0.01 | 10.95 | 54.41(1.02) |
| LMS | 0.006 | 0.004 | 0.001 | 0.022 | 0.044 | 0.053 | 0.001 | 0.168 | 0.169 | 2.465 | 0.379 | 0.02 | 0.001 | 0.002 | 0.098 | 0.026 | 0.02 | 3.484 | 70.76(1.65) |
| SVG | 0.064 | 0.01 | 0.008 | 0.006 | 0.007 | 0.01 | 0.001 | 0.397 | 0.462 | 0.025 | 2.71 | 0.012 | 0 | 0.013 | 0.643 | 0.002 | 0.024 | 4.399 | 61.6(1.57) |
| IWL | 0.01 | 0.002 | 0.044 | 0.029 | 0.103 | 0.022 | 0.002 | 0.048 | 0.017 | 0.008 | 0.042 | 2.673 | 0.024 | 0.001 | 0.012 | 0.224 | 0 | 3.263 | **81.91(1.45)** |
| CWL | 0.004 | 0.002 | 0.008 | 0.002 | 0.004 | 0.002 | 0.004 | 0.008 | 0.006 | 0 | 0.008 | 0.188 | 1.476 | 0.007 | 0.007 | 0.059 | 0 | 1.783 | **82.77(1.92)** |
| IMP | 0.074 | 0.011 | 0.008 | 0.008 | 0.037 | 0.002 | 0 | 0.041 | 0.024 | 0.002 | 0.014 | 0.004 | 0 | 5.087 | 0.01 | 0.004 | 0 | 5.329 | 95.45(0.61) |
| BAL | 0.048 | 0.01 | 0.002 | 0.004 | 0.002 | 0.001 | 0 | 0.193 | 0.328 | 0.557 | 0.582 | 0.043 | 0.002 | 0.035 | 5.384 | 0.029 | 0.108 | 7.33 | 73.45(1.11) |
| WTR | 0.014 | 0.024 | 0.014 | 0.014 | 0.019 | 0.008 | 0.006 | 0.011 | 0.016 | 0.007 | 0.011 | 0.168 | 0.114 | 0.011 | 0.019 | 3.054 | 0.002 | 3.509 | 87.04(1.22) |
| PSI | 0 | 0 | 0 | 0.001 | 0.002 | 0 | 0 | 0.005 | 0.03 | 0.001 | 0.011 | 0 | 0 | 0 | 0.019 | 0.023 | 1.363 | 1.455 | 93.65(1.37) |
| Total | 14.757 | 3.224 | 10.753 | 10.56 | 7.833 | 3.724 | 1.97 | 8.711 | 8.883 | 3.179 | 5.353 | 3.927 | 1.668 | 5.497 | 6.526 | 3.482 | 1.532 | | |
| U.A. (SE) | 82.85 | **59.92** | 84.55 | 64.76 | 72.52 | 49.77 | **39.34** | 62.44 | 67.07 | 77.55 | **50.63** | **68.07** | **88.49** | 92.54 | 82.5 | 87.73 | 88.96 | | |
| | (0.67) | (1.85) | (1.75) | (1) | (1.08) | (1.76) | (1.38) | (1.11) | (1.07) | (1.59) | (1.47) | (1.6) | (1.68) | (0.76) | (1.01) | (1.19) | (1.72) | | |
| O.A. | | | | | | | | 73.04% (±0.30%) | | | | | | | | | | | |

11. Results and discussion are very thorough although there is no mention to coastal regions which will be areas of major change detectable at 30 m resolution.

Great thanks for the comment. Yes, we completely agree that the coastal regions experienced obvious land-cover changes. To intuitively understand these coastal changes, an example in Yellow River Estuary Delta was also added in the Figure 8 and the corresponding descriptions as:

Lastly, the Yellow River Delta, as one of the typical coastal regions, was selected to understand the GLC_FCS30D for capturing these coastal land-cover changes. Obviously, the land-cover changes in the GLC_FCS30D can be concluded into three aspects: 1) a large amount of flooded flats and flat flats were reclaimed as the aquaculture ponds, especially after 2000; 2) the mouth of the Yellow River turned from south to north (black rectangle), that is, there were large land-cover changes between tidal/flooded flats, water bodies and salt marshes; 3) a lot of impervious surfaces encroached the coastal water-bodies and flats. In short, if we combine real time-series remote-sensing observation data, the GLC_FCS30D effectively captures the spatiotemporal changes of the land surface.

[Figure]

**Figure 8**. Three typical enlargements of land-cover changes in the GLC_FCS30D from 1985 to 2022 in (a) the Amazon rainforest, (b) the Yangtze River Delta in China, **and (c) the Yellow River Delta in China**. The color-coded legend is like the global map in Figure 4. In each case, the natural-color imagery from 1985 to 2022 is a composite taken from Landsat imagery.

12. Overall, I enjoyed reading this paper. The analysis was very thorough and easy to follow.

Great thanks for the positive comments. The analysis has been further improved based on your and another reviewer's comments.

---

## Author Response (AR2)

Dear Topical Editor and Reviewer:

On behalf of my co-authors, we thank you very much for reviewing our manuscript and giving us a lot of useful comments and suggestions. We appreciate the comments on our manuscript entitled "GLC_FCS30D: The first global 30-m land-cover dynamic monitoring product with a fine classification system from 1985 to 2022 using dense time-series Landsat imagery and continuous change-detection method" (essd-2023-320).

We have revised the manuscript carefully according to the comments. All the changes were high-lighted (red color) in the manuscript. And the point-by-point response to the comments of the reviewers is also listed below.

Looking forward to hearing from you soon.

Best regards,

Prof. Liangyun Liu

liuly@radi.ac.cn

Institute of Remote Sensing and Digital Earth, Chinese Academy of Sciences

No.9 Dengzhuang South Road, Haidian District, Beijing 100094, China

Comments for essd-2023-320

The manuscript focused on the effort of mapping global 30-m land cover and change in 1985-2022. An advanced continuous change detection and classification approach was chosen with Landsat imagery to characterize global 35 different land cover types. Also, independent reference dataset was collected to validation the mapping product. Other third-party validation datasets were also implemented to provide additional validations. Relatively high mapping accuracy was achieved from the mapping products. The basic idea and approach are interesting, and the data presented by the authors have good potential for an interesting paper. However, there are gaps and lack of clarity in some parts of the manuscript, in particular method section, followed by the introduction and result.

Great thanks for the comments. The manuscript has been improved according to your comments and suggestions.

**Major comments:**

1. Using data from stable areas as training dataset is a good way. However, this study used the mapped data itself in one year as the training source. It is not clear for me is that the stable area is just from 2020 detection or from all year detection. Or in other way, how these stable areas were determined need additional clarification.

Great thanks for the comment. The quality of training samples is the key in land-cover mapping and change monitoring. In this study, all training samples are derived from temporal stable areas during 1985-2022 instead of the single year of 2020. The stable areas were determined by the continuous change detection (CCD) algorithm in Section 3.2. More clarification was added to describe how the training samples were derived. The manuscript has been strengthened in Section 3.3.1 as:

"we combined the GLC_FCS30-2020 prior dataset and **the change-detection mask (derived from the CCD algorithm described in Section 3.2)** to obtain the spatiotemporally stable training samples. Specifically, temporally stable areas are known to have higher mapping accuracy (Yang and Huang, 2021; Zhang and Roy, 2017; Zhang et al., 2023); thus, **we first used the aforementioned CCD mask to retain these temporally stable areas during 1985-2022, and then overlap them into the GLC_FCS30-2020 maps to determine their land-cover labels**. Next, Radoux et al. (2014) emphasized that land-cover transition areas usually were subject to more serious misclassification problems and that the homogeneous land-cover pixels had a higher probability of achieving acceptable accuracy. Therefore, we used the morphological erosion filter of 3 pixels × 3 pixels to refine these temporally stable areas into spatiotemporally homogeneous areas."

2. The data in the stable areas do not guarantee the land cover types in these area are correct. More discussions are needed to justify the rational of this training data selection.

Yes, we agree that the training samples derived from the stable areas cannot be guaranteed to be completely correct. In this study, except for the temporally stable constraint, the spatial homogeneity checking (the morphological erosion filter of 3 pixels × 3 pixels) also applied to optimize the training samples. Next, the GLC_FCS30 in 2020 was demonstrated to achieve an overall accuracy of 82.5%, and showed obvious advantages in mapping accuracy and diversity of land-cover types comparing with other land-cover products. In addition, our previous studies in generating the GLC_FCS30 land-cover maps have demonstrated that the local adaptive random forest classification models also showed great robustness to the erroneous training samples as Figure S1 (Zhang et al., 2021):

[Figure]

Figure S1. Sensitivity analysis showing the relations between the overall classification accuracy at two different classification system and the percentage of total samples and erroneous sample points. The Figure S1 came from the work of Zhang et al. (2021).

The quantitative relationship in Figure S1 indicated that overall accuracy of two classification systems (level-0 and LCCS level-1) generally decreased with the increasing of percentage of erroneous sample points. It remained relatively stable when the percentage of erroneous training sample was controlled within 30%, and decreased obviously after exceeding the threshold of 30%.

Zhang, X., Liu, L., Chen, X., Gao, Y., Xie, S., and Mi, J.: GLC_FCS30: global land-cover product with fine classification system at 30 m using time-series Landsat imagery, Earth Syst. Sci. Data, 13, 2753-2776, https://doi.org/10.5194/essd-13-2753-2021, 2021.

In the revised manuscript, the discussion about the quality of derived training samples have been added in the training sample Section as:

Benefiting from the temporally stable checking during 1985-2022, spatial homogeneity analyzing, and the overall accuracy of 82.5% in GLC_FCS30-2020 products, these spatiotemporally stable areas are retained to generate the training samples. It should be noted that the spatiotemporally stable areas are not guaranteed to be completely accurate, that is, a small number of derived training samples might be mis-labeled. Fortunately, previous studies in large-area land-cover mapping demonstrated that the random forest classification models (adopted by this study in Section 3.3.2) is highly robust to the erroneous training samples (Gong et al., 2019b; Mellor et al., 2015; Zhang et al., 2021b). For example, Gong et al. (2019b) found that the overall accuracy kept relatively stable when the proportion of erroneous training samples is controlled within 20%. Thus, the used spatiotemporally stable areas can be supported to derive confident training samples and further ensure the quality of land-cover dynamic monitoring.

3. A 26-time steps were used to map land cover by every five years before 2000 and annually after 2000 (Page 3, lines 119-120). However, why such steps are used did not given. The authors need to explain because you may mapped annual land cover and change in these periods without having too much extra cost.

Great thanks for the comment. Yes, we agree that we can generate the annual land-cover change maps during before 2000 without too much extra cost. However, why we still choose to update land-cover changes with 5-years interval before 2000 are the Landsat observations before 1999 only came from the single Landsat 5 TM sensor, which meant the valid observation imagery are too sparse. As we all known, the number of observation

data greatly affected the land-cover mapping and change detection, thus, we **have to sacrifice the temporal cycle (from one year to 5-years) to improve the land-cover change monitoring quality**. In this study, we combine the satellite observations from two years before and after the nominal center year from 1985 to 1995; for example, we update the land-cover maps in 1995 using all available imagery from 1993 to 1997.

The reasons why we choose the GLC_FCS30D updated every 5 years before 2000 has been added in the Introduction Section as:

In this study, we had the following three aims: 1) use the continuous change-detection algorithm and full time-series Landsat observations to generate the first global 30-m land-cover dynamic products with fine classification system (GLC_FCS30D) from 1985 to 2022, which contains 35 fine land-cover subcategories with 26 time-steps (maps updating every five years before 2000 and annually after 2000). **It should be noted that the GLC_FCS30D updated every five-years before 2000 due to the sparse availability of Landsat 5 imagery, thus, we combine the satellite observations from two years before and after the nominal center year from 1985 to 1995 for ensuring the mapping accuracy of GLC_FCS30D before 2000.**

In the Method Section, the reason is also been added as:

Last, since the land-cover distribution was usually related to the topographical environment, for example, croplands and water bodies are mainly distributed in flat areas, three topographical variables (elevation, slope, and aspect), calculated from a global 30 m DEM dataset (named as: ASTER_GDEM) (Tachikawa et al., 2011), were also imported. **In addition, due to the limited storage capacity and satellite–ground data-transmission capacity of early satellites, the density of Landsat imagery is sparse before 2000 (only Landsat 5 single-satellite acquired data) (Roy et al., 2014b). We choose the coarse temporal cycle of 5-years for ensuring the mapping accuracy before 2000, that is, the satellite observations from two years before and after was used for the nominal center year. For example, we update the land-cover maps in 1995 using all available imagery from 1993 to 1997.** In total, there were 49 multisource features, including 40 phenological spectra features, 6 texture features, and 3 topographical variables.

Meanwhile, the Section 4.5 also explained that one of our further works would combine multisourced remote sensing imagery to achieve the goal of global annual land-cover change monitoring before 1985 as:

Additionally, due to the limited storage capacity and satellite–ground data-transmission capacity of early satellites, the density of Landsat imagery is sparse before 2000 (only Landsat 5 single-satellite acquired data) (Roy et al., 2014b). In this study, we combine the satellite observations from two years before and after the nominal center year from 1985 to 1995; for example, we update the land-cover maps in 1995 using all available imagery from 1993 to 1997. However, a previous study found that northeastern Asia did not have any valid Landsat observations before 2000 (Zhang et al., 2022), which means some land-cover changes could not be captured in the GLC_FCS30D in these areas before 2000. To solve the problem of missing and sparse observations, a useful solution is to fuse multisourced remote-sensing imagery. For example, Zhang et al. (2021c) combined Landsat and Sentinel-2 imagery to track tropical forest disturbances with overall accuracy of more than 87%. Therefore, further work will investigate the feasibility of integrating Sentinel 1/2, SPOT, MODIS, and AVHRR imagery as auxiliary datasets to achieve the annual land-cover monitoring before 2000 and further ensure the land-cover monitoring quality.

The Section 5 "Data availability" also added the description about the different updating cycle as:

The developed GLC_FCS30D dataset can be freely accessible via https://doi.org/10.5281/zenodo.8239305 (Liu et al., 2023). To allow users to better select this dataset, it is saved as 961 5° × 5° independent tiles. Each tile is named as 'GLC_FCS30D_yyyyYYYY_E/W**N/S**.tif', in which 'E/W**N/S**' represents the longitude and latitude coordinates of the top-left corner, and yyyy and YYYY are the start and end years of the land-cover

change monitoring. The GLC_FCS30D contains 26 time-step maps from 1985 to 2022, updated every five years before 2000 and annually from 2000 to 2022. **It should be noted that the GLC_FCS30D adopted the 5-years cycle before 2000 because of the sparse availability of Landsat 5 imagery at early stage, thus, we sacrificed temporal cycle in guaranteeing the land-cover mapping accuracy.**

4. A moving window of 3x3x3 are used to optimize temperature consistency of mapped land cover. However, such smooth window approach could also remove some rare classes that may not be larger than these moving windows. Some sensitivity analysis should be performed to compare the classification with and without moving window.

Great thanks for the comment. Yes, the temporal consistency optimization using the moving window of 3x3x3 maybe smooth some rare classes. In most case, the rare land-cover types, which are smaller than the windows, still can be retained because of the empirical of 0.5. It should be noted that the method was firstly designed for optimizing the impervious surfaces in the work of Li et al.,(2014), (the impervious surfaces are the representative rare land-cover type), and their analysis showed that the optimization method still retained most small rural impervious surfaces.

Li, X., Gong, P., and Liang, L.: A 30-year (1984–2013) record of annual urban dynamics of Beijing City derived from Landsat data, Remote Sensing of Environment, 166, 78-90, https://doi.org/10.1016/j.rse.2015.06.007, 2015.

In this revision, we also analyzed performance of the optimization method at two typical tiles (Amazon's deforestation and urban expansion in Yangtze River Delta) with or without the moving window as:

[Figure]

[Figure]

| Without optimization in 2000 | **With optimization** in 2000 | Without optimization in 2022 | **With optimization** in 2022 |

Figure S2. The comparisons before and after temporal consistency optimization in the Amazon's deforestation areas and urban rapid expansion area and their randomly enlargements.

Intuitively, the temporal consistency optimization preserved the spatial details well, that is, most small or fragmented forests and impervious surfaces were retained completely. Meanwhile, this post-processing showed great ability to deal with the 'salt and pepper' noise caused by the pixel-based classification, especially in the R1 and R3, namely, the optimized maps were 'visually-clear' than the origin maps

In summary, the comparisons indicates that the temporal optimization still retain the rare land-cover types and further improve the land-cover temporal consistency by removing some 'salt and pepper' noise caused by the pixel-based classification. It should be noted that this temporal-consistency optimization algorithm attached wide attentions in land-cover post-processing, for example, Yang et al., (2021) applied the same method to optimize the China annual land-cover products (CLCD) during 1990-2019, and Xie et al. (2021) used the similar method to improve the quality of land-cover change monitoring in Beijing during 2000-2021.

Yang, J. and Huang, X.: The 30 m annual land cover dataset and its dynamics in China from 1990 to 2019, Earth Syst. Sci. Data, 13, 3907-3925, https://doi.org/10.5194/essd-13-3907-2021, 2021.

Xie, S., Liu, L., Zhang, X., and Yang, J.: Mapping the annual dynamics of land cover in Beijing from 2001 to 2020 using Landsat dense time series stack, ISPRS Journal of Photogrammetry and Remote Sensing, 185, 201-218, https://doi.org/10.1016/j.isprsjprs.2022.01.014, 2022.

In the manuscript, the discussion about the temporal consistency optimization was also added in the Section 4.5 as:

To ensure stability of the GLC_FCS30D, a spatiotemporal consistency optimization algorithm, which was widely used in impervious surface change optimizations (Li et al., 2015; Zhang et al., 2022), was applied. It makes full use of the spatiotemporal neighbor pixels to calculate the land-cover homogeneity, and then remove the 'salt and pepper' noise caused by the pixel-based classifications. The qualitative comparisons in Amazon's deforestation areas and China's urban expansion areas (Figure S2 in the supplement material) also showed that the spatiotemporal consistency optimization can improve the data quality of GLC_FCS30D by suppressing 'salt and pepper' noise and optimizing the temporal consistency. Similarly, Yang and Huang (2021) used this algorithm to optimize China's annual land-cover products during 1999 during 1990-2019, and found that it improved the mapping accuracy of the time-series land-cover dataset.

5. Figure 5 (b) and Figure 6 explain the global land cover change and changes in different regions. A global land cover spatial distribution in 1985 should be included so that reader can see the difference from 2022 map listed in Figure 4.

Great thanks for the suggestion. The GLC_FCS30D in 1985 has been added into the Supplement material:

[Figure]

**Figure S3**. The overview of the GLC_FCS30D land-cover maps in 1985 and 2022.

**Detail comments:**

1. Page 1, line 21. "GLC-FCS30D is described as the first global 30-m land-cover dynamic monitoring dataset," The statement is not true. Several global 30-m land cover data, Copernicus Global land Service (100 m) 2015-2019, China Globeland30 (2000 and 2010 in 30-m), 9-class annual land use and land cover by Impact Observatory, are also available.

Great thanks for your comment. The statement has been revised as "GLC_FCS30D is described as the novel global 30-m fine land-cover dynamic monitoring dataset".

The similar statement on Page 2, line 75-79 needs to be changed to include the current existed global land cover products.

Based on your suggestion, the CGLC100 and GlobeLand30 land-cover products have been added into the Introduction Section.

In the early stage, GLCCM mainly relied on the time-series MODIS, AVHRR, and Project for Onboard Autonomy (PROBA)-V imagery (**Buchhorn et al., 2020**; Friedl et al., 2010); for example, Sulla-Menashe et al. (2019) generated a global 500-m annual land-cover products (MCD12Q1) from 2001 to present using time-series MODIS imagery with an overall accuracy of 73.6%.

Recently, benefitting from the free access to fine-resolution satellite imagery and powerful computing and storage capabilities, especially after the rise of cloud computing [such as Google Earth Engine (Gorelick et al., 2017) and Microsoft Planetary Computer], fine-resolution land-cover dynamic monitoring is experiencing rapid development. Correspondingly, numerous national and global 30-m land-cover dynamic products have been developed (**Chen et al., 2015**; Homer et al., 2020; Liu et al., 2021a; Potapov et al., 2022; Yang and Huang, 2021; Zhang et al., 2022).

Buchhorn, M., Lesiv, M., Tsendbazar, N.-E., Herold, M., Bertels, L., and Smets, B.: Copernicus Global Land Cover Layers—Collection 2, Remote Sensing, 12, 1044, https://doi.org/10.3390/rs12061044, 2020.

Chen, J., Chen, J., Liao, A., Cao, X., Chen, L., Chen, X., He, C., Han, G., Peng, S., Lu, M., Zhang, W., Tong, X., and Mills, J.: Global land cover mapping at 30m resolution: A POK-based operational approach, ISPRS Journal of Photogrammetry and Remote Sensing, 103, 7-27, https://doi.org/10.1016/j.isprsjprs.2014.09.002, 2015.

2. Page 4, lines 135-136. Why only listed the spectral difference between ETM+ and OLI?

Thanks for the comment. The previous studies have explained that the TM and ETM+ shared the same spectral response function, and the spectral difference between TM and ETM+ are directly ignored (Roy et al. 2016). The make the sentence clearer, it has been revised as:

Then, although the Landsat 5, 7, 8, and 9 missions share similar spectral bands, the wavelength differences between the **TM,** ETM+ and OLI cannot be ignored. Relative radiometric normalization was applied to the TM and ETM+ imagery using the transformation coefficients suggested by Roy et al. (2016).

3. Page 11, lines 307-308/ sample size of 8000 and 600 are recommended. Are these sample thresholds for one tile?

Thanks for the comment. Yes, the recommended sample size of 8000 and 600 are suitable for all 961 5° × 5° geographical tiles. The corresponding explanation has been revised as:

Thus, the maximum and minimum sample size for abundant and rare land-cover types were suggested as **8000 and 600**, came from the study in Zhu et al. (2016), for avoiding the extremes of sample sizes. Next, the GLC_FCS30-2020 products were split into 961 5° × 5° geographical tiles, and we used the areal-proportional sampling strategy and **two sample balancing thresholds (8000 and 600 for maximum and minimum sample size)** to allocate the training samples from the spatiotemporally stable areas **in each 5° × 5° geographical tile**. Last, the impervious surface and wetland samples were excluded because both have been independently developed as the thematic datasets in Section 2.3 and 2.4.

4. Page 14, lines 400 -401. What are 10 major land cover types?

Great thanks for the comment. The 10 major land-cover types have been explained in the Table 1, which include **cropland, forest, shrubland, grassland, tundra, wetland, impervious surface, bare areas, water body,**

**permanent snow and ice**. To make the sentence clearer, it has been revised as:

The GLC_FCS30D adopts a fine classification system containing 35 subcategories, for which we applied an analysis protocol into the basic classification system and the LCCS level-1 validation system, **whose details were explained in the Table 1 and contained 10 major land-cover types and 17 fine land-cover types (Table 1)**, respectively.

5.  Figure 5a. The graphic is good to show the spatial distributions of land cover change intensity. However, there are no further discussions about these changes, especially for some area experiencing very change high intensity, e.g., changes in the southern Hudson Bay area in Canada, western coast of African, and eastern side of Australia.

Great thanks for the comment. Actually, these areas experienced very change high intensity have been independently discussed as:

"Obviously, global land-cover has experienced significant changes over the past 37 years, mainly in the following three typical areas: 1) tropical rainforest peripheral areas in South America and Southeast Asia, in which deforestation is the dominant cause; 2) wetland and water-body intermingling areas, such as **North America** and northern Asia, i**n which water bodies and wetland were transformed into one another due to different annual water levels. In the GLC_FCS30D the water body land-cover type represents permanent water during the year (it may be wetland in other years)**. 3) The semi-arid areas in **Australia**, Central Asia, and **western Africa**, where land cover (such as sparse vegetation or bare land) is directly affected by precipitation and temperature. For example, if there is sufficient precipitation in the year, the sparse vegetation and some bare land would be covered by grass in semi-arid areas. Similarly, the work of Winkler et al. (2021) revealed that these semi-arid areas experienced serious and frequent land-cover changes."

Namely, the changes in the southern Hudson Bay area in Canada are obvious because the water bodies and wetland were transformed into one another due to different annual water levels. As for the western coast of African and eastern side of Australia, both of them belongs to the semi-arid areas, where land cover (such as sparse vegetation or bare land) is directly affected by precipitation and temperature. For example, if there is sufficient precipitation in the year, the sparse vegetation and some bare land would be covered by grass in semi-arid areas.

6.  Full land cover names for these synonyms names in Table 2 & 3 need to be listed.

Great thanks for the comment. The full land-cover names of these synonyms names have been moved into the Table 1 based on the Reviewer 2 suggestions in the first round. In this revised manuscript, the notes about the synonyms have been added in the **lower left corner of Table.**

7.  Figure 10. What happened for the UA of bare land in 2005-2015? What caused such sudden increase in UA?

Great thanks for the comment. The sudden increase of U.A. in Figure 10 is Ice and snow (instead of bare land). The cause is that the Ice and snow is a greatly sparse land-cover type in the U.S., its validation sample size in the LCMAP is also small and a little misclassified grass/bare land pixels are correctly identified as snow and ice during 2005-2014. The manuscript has been revised as:

The large fluctuations in U.A. for ice and snow are attributed to: 1) the small size of ice and snow samples in the LCMAP_Val dataset, 2) a few misclassified grass/bare land pixels correctly identified as snow and ice during 2005-2014.